



# Ocean acidification changes the structure of an Antarctic coastal protistan community

Alyce M. Hancock[1,2,3], Andrew T. Davidson[3,4], John McKinlay[4], Andrew McMinn[1,2,3], Kai Schulz[5], and Rick L. van den Enden[4]

[1]Institute of Marine and Antarctic Studies, University of Tasmania 20 Castray Esplanade, Battery Point TAS Australia 7004
[2]Antarctic Gateway Partnership, 20 Castray Esplanade, Battery Point TAS Australia 7004
[3]Antarctic Climate & Ecosystems Cooperative Research Centre 20 Castray Esplanade, Battery Point TAS Australia 7004
[4]Australian Antarctic Division 203 Channel Hwy, Kingston TAS Australia 7050
[5]Centre for Coastal Biogeochemistry Research, Southern Cross University

*Correspondence to:* Alyce Hancock (alyce.hancock@utas.edu.au)

**Abstract.** Antarctic near-shore waters are amongst of the most vulnerable in the world to ocean acidification. Microbes occupying these waters are critical drivers of ecosystem productivity, elemental cycling and ocean biogeochemistry, yet little is known about their sensitivity to ocean acidification. An unreplicated, six-level dose-response experiment was conducted using 650 L incubation tanks (minicosms) adjusted to fugacity of carbon dioxide ($f\mathrm{CO}_2$) from 343 to 11641 µatm. The minicosms were

filled with near-shore water from Prydz Bay, East Antarctica and the protistan composition and abundance was determined by microscopy analysis of samples collected during the 18 day incubation. No $\mathrm{CO}_2$-related change in the protistan community composition was observed during the initial 8 day acclimation period under low light. Thereafter, the response of protists to $f\mathrm{CO}_2$ were species-specific for both heterotrophic and autotrophic protists. The response by diatoms was related to cell size, large cells increasing in abundance with low to moderate $f\mathrm{CO}_2$ (634-953 µatm). Similarly, the abundance of *Phaeocystis*

*antarctica* increased with increasing $f\mathrm{CO}_2$ peaking at a $f\mathrm{CO}_2$ of 634 µatm. Above this threshold the abundances of large diatoms and *Phaeocystis antarctica* fell dramatically, and small diatoms dominated, therefore culminating in a significant shift in the composition of the protistan community. The threshold $\mathrm{CO}_2$ level at which the composition changed agreed with that previously measured at this location, indicating it remains consistent among seasons. This suggests that near-shore microbial communities are likely to change significantly near the end of this century if anthropogenic $\mathrm{CO}_2$ release continues unabated,

with profound ramifications for near-shore Antarctic ecosystems.

# 1 Introduction

Eukaryotic and prokaryotic microbes are the most abundant organisms in the oceans and comprise the base of all marine foodwebs (Kirchman, 2008; Cooley and Doney, 2009; Doney et al., 2012). Their composition and abundance determines the



quality and quantity of food available to higher trophic levels, affecting fishery productivity and the conservation of biological diversity (Cooley and Doney, 2009; Doney et al., 2012). In the Southern Ocean, microbes are drivers of productivity, elemental cycles and ocean biogeochemistry, meaning their response to environmental stressors is a key determinant of Southern Ocean feedbacks to global climate change (Arrigo and Thomas, 2004; Arrigo et al., 2008; Kirchman, 2008). Despite their importance,

relatively little is known about the sensitivity of Antarctica marine microbes to ocean acidification and this limits our ability to predict how the Southern Ocean will be impacted in the future, and the feedback this may have on global climate change.

The Southern Ocean is particularly vulnerable to ocean acidification due to its cold temperatures, extensive upwelling and naturally large seasonal fluctuations in pH. Carbon dioxide ($CO_2$) has a higher solubility at cold temperatures so that more $CO_2$ is being absorbed in polar waters compared to that in warmer waters. Added to this, the surface waters of the Southern

Ocean are being exposed to increased $CO_2$ from upwelling bringing deep $CO_2$ rich waters to the surface (Orr et al., 2005). This is enhanced in near-shore Antarctica where upwelling is greater (McNeil et al., 2010; IPCC, 2011). These waters already experience large seasonal fluctuations in pH, upon which the anthropogenic decrease is superimposed. Prydz Bay, off Davis Station East Antarctica, shows an annual cycle of $CO_2$ levels ranging from as high as 450 µatm during autumn and winter to below 100 µatm in summer (Gibson and Trull, 1999; Roden et al., 2013). During autumn and winter, sea-ice covers the

ocean, light is limiting to photosynthesis, the draw-down of $CO_2$ by primary production and air-sea gas exchange is negligible. During spring and summer the sea-ice retreats and phytoplankton bloom, due to the increased light and nutrient availability (Gibson and Trull, 1999; Roden et al., 2013). Thus, phytoplankton in these coastal Antarctic waters are exposed to highly variable carbonate chemistry over the annual cycle.

There have been relatively few studies on the response of microbial communities to increased $CO_2$ in the Southern Ocean

and Antarctic waters, and out of these fewer have been at the community level (Tortell et al., 2008; Feng et al., 2010; McMinn et al., 2014; Coad et al., 2016; Davidson et al., 2016; Tarling et al., 2016; Thomson et al., 2016; McMinn et al., 2017). These studies report a range of microbial responses to increased $CO_2$, but one common trend was a shift in community composition. With increased $CO_2$ Tortell et al. (2008) and Feng et al. (2010) found an increase in the dominance of *Chaetoceros* species over smaller, pennate diatoms such as *Pseudonitzschia subcurvata*. Tortell et al. (2008) also found an increase in *Phaeocystis*

*antarctica* abundance at 800 µatm, although Feng et al. (2010) found no effect on the same species. Davidson et al. (2016) and Thomson et al. (2016) ran a minicosm study at the same location as this study (Davis Station, East Antarctica), and found a significant change in species abundance and biomass with increases in $CO_2$ above 643 to 1281 µatm (Davidson et al., 2016; Thomson et al., 2016). Below this threshold the community had a high abundance of large diatoms, but above was dominated by nano- and picoplankton and *Fragilariopsis curta/cylindrus* <10 µm long. This is consistent with microbial communities

studied elsewhere in the world, particularly in the Arctic, where studies have observed a shift to pico- and nanoplankton at high $CO_2$ (Hare et al., 2007; Brussaard et al., 2013). Schulz et al. (2017) recently reviewed 31 community level studies, finding an increase in picoeukaryotes at high $CO_2$ in most studies, particularly prasinophytes and chlorophytes. The effects of increased $CO_2$ on larger marine diatoms was less clear with evidence for both promotion and inhibition. Schulz et al. (2017) concluded that the effects on marine diatoms are likely to be at a species level rather than the community level, and therefore could be

more difficult to detect.



Incubations of natural communities, which include the effects of species interactions and competition, are essential to predict the effects of ocean acidification on microbial communities (Schulz et al., 2017). This study will assess the following questions on a natural microbial community from near-shore East Antarctic waters.

1. Do individual species have different tolerances to increased $CO_2$ level?

2. Does the protistan community composition and abundance change with increased levels of $CO_2$? And what $CO_2$ level elicits this change?

3. Does an acclimation period allow the protistan community to better tolerate exposure to elevated $CO_2$?

4. When compared with Davidson et al. (2016) and Thomson et al. (2016), does our experiment indicate that the response by the protistan community at this site is consistent in nature and threshold, irrespective of seasonal and interannual
differences in the composition of the community and the availability of nutrients?

## 2  Methods

An unreplicated, six-level dose-response experiment was conducted on a natural microbial community over a range of $CO_2$ levels (343, 506, 634, 953, 1140 and 1641 µatm). Seawater was collected on the 19th November 2014 approximately 1 km offshore from Davis Station, Antarctica (68°C 35' S, 77°C 58' E) from an area of ice-free water amongst broken fast-ice. The
seawater was collected using a thoroughly rinsed 720 L Bambi bucket slung beneath a helicopter and transferred into a 7000 L polypropalene reservoir tank. Six 650 L polyethene tanks (minicosms), located in a temperature-controlled shipping container, were immediately filled via teflon lined hose via gravity with an in-line 200 µm Arkal filter to exclude metazooplankton. The minicosms were simultaneously filled to ensure they contained the same starting community. The ambient water temperature at time of collection was -1.0°C and the minicosms were maintained at a temperature of 0°C ± 0.5°C. At the centre of each
minicosm there was an auger shielded for much of its length by a tube of polythene. This auger was rotated at 15 rpm to gently mix the contents of the tanks. Each minicosm tank was covered with an acrylic air-tight lid to prevent $CO_2$ off-gassing outside of the minicosm headspace. For a more detailed description of the minicosm set-up see Davidson et al. (2016).

The minicosm experiment was conducted between the 19th November and the 7th December 2014. Initially, the contents of the tanks were given a day to equilibrate to the minicosms. This was followed by a five day acclimation period to increasing
$CO_2$ at low light ($0.8 \pm 0.2 \, \mu mol \, m^{-2} \, s^{-1}$), allowing cell physiology to acclimated to the $CO_2$ increase (days 1-5). During this period the $CO_2$ was progressively adjusted over five days to the target level for each tank (343-1641 µatm). Thereafter $CO_2$ was adjusted daily to maintain the $CO_2$ level in each treatment (see carbonate chemistry section below). Following acclimation to the various $CO_2$ treatments, light was progressively adjusted to $89 \pm 16 \, \mu mol \, m^{-2} \, s^{-1}$ at a 19 h light:5 h dark cycle. The community was incubated and allowed to grow for a further 10 days (days 8-18) with $CO_2$ adjusted back to target each day (see
carbonate chemistry section below). For a detailed description of lighting apparatus and climates see Davidson et al. (2016) and Deppeler et al. (submitted).



## 2.1 Carbonate chemistry measurements and calculations

Carbonate chemistry was calculated from dissolved inorganic carbon (DIC), $pH_T$ (total proton scale), salinity and temperature. Minicosm fugacity of $CO_2$ ($fCO_2$) was adjusted incrementally during the acclimation phase to the target level for each tank, and then for the remainder of the experiment $fCO_2$ was adjusted daily to maintain the target levels within the tanks. Daily adjustments were made based on morning and afternoon pH measurements, and were achieved by the adding the required volume of $0.2\,\mu m$ filtered $CO_2$ saturated seawater. After adjustment, full carbonate chemistry measurements were performed (DIC, pH, salinity and temperature) to check the adjustment, and if required a secondary adjustment was made. Daily pH measurements were conducted using a portable, NBS-calibrated probe (Mettler Toledo) and salinity measurements via a conductivity metre WTW197. DIC was measured by infra-red absorption on an Apollo SciTech AS-C3 analyzer equipped with a LICOR7000 and $pH_T$ with a GBC UV-Vis 916 spectrophotometer in a ten centimetre thermostated cuvettte and pH indicator dye (m-cresol purple). All samples for the carbonate chemistry were collected in $500\,mL$ glass bottles with stoppers following the Dickson et al. (2007) guidelines. Full carbonate chemistry measurements and calculations are described in Deppeler et al. (submitted).

## 2.2 Microbial community structure

Samples from each minicosm were collected on days 1, 3, 5, 8, 10, 12, 14, 16 and 18 for microscopic analysis to determine protistan identity and abundance. Approximately $960\,mL$ were collected from each tank, on each day. Samples were fixed with $40\,mL$ of Lugol's iodine and allowed to sediment out at $4\,°C$ for $\geq 4$ days. Once cells had settled the supernatant was gently aspirated till approximately $200\,mL$ remained. This was transferred to a $250\,mL$ measuring cylinder, again allowed to settle (as above), and the supernatant gently aspirated. The remaining ~$20\,mL$ was transferred into a $30\,mL$ amber glass bottle.

A further 1 L was taken on days 0, 6, 13 and 18 for analysis by Field Emission Scanning Electron Microscope (FESEM). These samples were concentrated to $5\,mL$ by filtration over a $0.8\,\mu m$ polycarbonate filter. Cells were resuspended, the concentrate transferred to a glass vial and fixed to a final concentration of 1% EM-grade gluteraldehyde (ProSciTech Pty Ltd).

All samples were stored and transported at $4\,°C$ to the Australian Antarctic Division, Hobart, Australia for analysis.

### 2.2.1 Electron microscopy

Gluteraldehyde-fixed samples were prepared for FESEM imaging using a modified polylysine technique (Marchant and Thomas, 1983). In brief, a few drops of gluteraldehyde-fixed sample were placed on polylysine coated cover slips and post-fixed with $OsO_4$ (4%) vapour for 30 min, allowing cells to settle onto the coverslips. The coverslips were then rinsed in distilled water and dehydrated through a graded ethanol series ending with emersion in 100% dry acetone before being critically point dried in a Tousimis Autosamdri-815 Critical Point Drier. The coverslips were mounted onto $12.5\,mm$ diameter aluminium stubs and sputter-coated with $7\,nm$ of platinum/palladium in a Cressington 208HRD coater. Imaging of stubs was conducted by JEOL JSM6701F FESEM and protists identified using Scott and Marchant (2005).




### 2.2.2 Light microscopy

Lugols-fixed and sedimented samples were analysed by light microscopy between July 2015 and February 2017. Between 2 to 10 mL (depending on cell-density) of lugols-concentrated samples was placed into a 10 mL Utermohl cylinder (Hydro-Bios, Keil) and the cells allowed to settle overnight. Due to the large variation in size and taxa, a stratified counting procedure was

employed to ensure both accurate identification of small cells and representative counts of larger cells. All cells greater than 20 µm were identified and counted at 20x magnification; those less than 20 µm at 40x magnification. For larger cells (>20 µm), 20 randomly chosen fields of view (FOV) at $3.66x10^6 \mu m^2$ were counted to gain an average cells per L. For smaller cells (<20 µm), 20 randomly chosen FOVs at $2.51x10^5 \mu m^2$ were counted. Counts were conducted on an Olympus IX 81 microscope with Nomarski interference optics. Identifications were determined using Scott and Marchant (2005) and FESEM images.

Autotrophic protists were distinguished from heterotrophs via the presence of chloroplasts and based on their taxonomic identity.

### 2.3 Statistical analysis

The minicosm experiment was a six dose, unreplicated experiment based on a repeated measures design. Due to the lack of replication, no formal statistics could be undertaken on the interactions between time and $f\mathrm{CO_2}$ treatment. Temporal changes

in species abundances between treatment groups were informally assessed by plotting the mean microbial abundance at each time for each treatment. Means and standard errors were calculated from separate FOV counts; as these are sub-samples are from a single treatment, they should be considered pseudo-replicates and are indicative of treatment-level sampling variability. Abundant taxa with low variance were examined separately, but rare taxa with high variance were combined into functional groups (see Table 1 for abbreviations). In plots, data points from the different $f\mathrm{CO_2}$ treatments were slightly offset at each

sample time to avoid over-plotting of the data. Cluster analyses and ordinations were performed on Bray-Curtis resemblance matrixes formed from square-root transformed abundance data. This transformation was assessed as appropriate for reducing the influence of abundant species, as judged from a one-to-one relationship between observed dissimilarities and ordination distances (ie. Shepard diagram, not shown). The Bray-Curtis metric was used as it is recommended for ecological data due to its treatment of joint absences (ie. these do not contribute towards similarity), and giving more weight to abundant taxa rather than

rare taxa (Bray and Curtis, 1957). The data days 1 to 8 and then days 8 to 18 were analysed separately to distinguish community structure in the acclimation period and in the exponential growth phase during the incubation period of the experiment.

Hierarchical agglomerative cluster analyses, based on the Bray-Curtis resemblance matrix, were performed using group-average linkage. Significantly different clusters of samples were determined using SIMPROF (similarity profile permutations method) (Clarke et al., 2008) with an alpha value of 0.05 and based on 1000 permutations. An unconstrained ordination by

non-metric multidimensional scaling (nMDS) was performed on the resemblance matrix with a primary ('weak') treatment of ties (Kruskal, 1964a, b). This was repeated over 50 random starts to ensure a globally optimal solution according to Peres-Neto and Jackson (2001). Clusters are displayed in the nMDS using colour. Weighted average of sample scores are shown in the





nMDS to show the approximate contribution of each species to each sample. The assumption of a linear trend for predictors within the ordination was checked for each covariate, and in all instances was found to be justified.

A constrained canonical analysis of principal coordinates (CAP) was conducted according to the Oksanen et al. (2017) protocol using the Bray-Curtis resemblance matrix. This analysis was used to assess the significance of the environmental covariates, or constraints, in determining the microbial community structure. Unlike the nMDS ordination, the CAP analysis uses the resemblance matrix to partition the total variance in the community composition into unconstrained and constrained components, with the latter comprising only the variation that can be attributed to the constraining variables, $f\mathrm{CO_2}$, Si, P and $\mathrm{NO_x}$. Random reassignment of sample resemblance was performed over 199 permutations to compute the pseudo-$F$ statistic as a measure of significance of each environmental constraint in the structural change of the microbial community (Legendre and Anderson, 1999). A forward selection strategy was used to choose a minimum subset of significant constraints that still account for the majority of the variation within the microbial community (Legendre et al., 2011).

All analyses were performed using R v1.0.136 (R Core Team, 2016) and the add-on package vegan v2.4-2 (Oksanen et al., 2017).

## 3 Results

### 3.1 Protistan community overview

The starting microbial community was characteristic of a post sea-ice break-out community in the near-shore seawater of Prydz Bay. It was highly diverse with over 100 species present, ranging from small flagellates ($<2\,\mu\mathrm{m}$) to large diatoms ($>100\,\mu\mathrm{m}$). The total protistan abundance at the beginning of the experiment was quite low (approximately $3\mathrm{x}10^5\,\mathrm{cells\,L^{-1}}$), but increased to between $6.4\mathrm{x}10^6$ and $1.9\mathrm{x}10^7\,\mathrm{cells\,L^{-1}}$ depending on the treatment. Abundances remained low during the acclimation period (days 1 to 8) then increased exponentially from days 10 to 16 in all treatments except the highest $f\mathrm{CO_2}$ treatment (Figure 1). At day 18 there was a decrease in abundance in all treatments except 635 and 1641 $\mu\mathrm{atm}$. The high variance in the 635 $\mu\mathrm{atm}$ treatment on day 18 is likely a due to the increase in a few rare large diatom species, which were highly variable. The highest $f\mathrm{CO_2}$ treatment showed a different trajectory to the other tanks, with low abundance maintained until day 12, after which cell numbers increased exponentially; hereafter this is referred to as a lag in growth.

### 3.2 Species specific $f\mathrm{CO_2}$ tolerances

#### 3.2.1 Diatoms

Diatoms dominated the microbial community and showed marked responses to increased $f\mathrm{CO_2}$ levels. The response of diatoms was size related, with small diatoms ($<20\,\mu\mathrm{m}$) showing little effect of exposure to higher $f\mathrm{CO_2}$, while larger diatoms ($>20\,\mu\mathrm{m}$) increased in abundance at moderate levels $f\mathrm{CO_2}$ ($\leq 634\,\mu\mathrm{atm}$) but declined at higher $f\mathrm{CO_2}$. This trend was particularly evident in discoid centric diatoms (centric diatoms of the genera *Thalassiosira*, *Landeria* and *Stellarima*), which showed decreased $f\mathrm{CO_2}$ tolerance with increasing size. The smallest discoid taxa, an unidentified 1 to 2 $\mu\mathrm{m}$ diameter centric diatom,



showed no significant response to increasing $f\text{CO}_2$ (Figure 2a). *Thalassiosira antarctica*, which ranged in size from 10 to 20 µm diameter, also showed no effect of increased $f\text{CO}_2$ until the highest treatment level of 1641 µatm (Figure 2b). *Landeria annulata*, a medium sized cell of 30 to 60 µm diameter, had an increases in abundance in low to moderate $f\text{CO}_2$ treatments (343-634 µatm) after which abundances decreased but not lower than that of the control treatment (Figure 2c). The two larger

dominant centric species, *Stellarima microtrias* (40 to 80 µm diameter) and *Thalassiosira ritscheri* ( >50 µm diameter) showed an increase in abundance with moderate increases in $f\text{CO}_2$ ($\leq$ 634 µatm), but they showed a decrease in abundance at the three higher $f\text{CO}_2$ levels (Figure 2d and e respectively). Whilst these two larger species had a similar response, there were subtle differences that correlate with differences in the average size of the species; *Stellarima* was slightly more tolerant than *T.ritscheri* (average sizes of *Stellarima* is less than *T.ritscheri*).

A similar size-related response was observed in *Fragilariopsis* cells (mostly *F.cylindrus* but included occasional *F.curta* and *F.kerguelensis*). *Fragilariopsis* was the dominant diatom, comprising between 15 to 50% of the total phytoplankton abundance and ranged in length from 2 µm to >50 µm. Toward the end of the incubation (days 16 and 18), the abundance of small *Fragilariopsis* cells (<20 µm) was higher in treatments exposed to moderately enhanced $f\text{CO}_2$ (506-953 µatm) compared to that of the control. Abundances in the $f\text{CO}_2$ treatments >953 µatm were lower but less than those in the control treatment

(Figure 3a). In contrast, larger cells of *Fragilariopsis* (>20 µm), showed a lower tolerance with abundances highest in the three lower $f\text{CO}_2$ treatments and considerably lower abundances in the two highest $f\text{CO}_2$ treatments. The abundance of large cells in the 953 µm treatment fell between these two extremes (Figure 3b).

Other diatoms showed similar responses with larger species having higher abundances in $f\text{CO}_2$ treatments $\leq$634 µatm but lower abundances in the three highest $f\text{CO}_2$ treatments ($\geq$953 µatm). This included members of the centric diatom

genus *Odontella* (mainly *O.weissfloggi* but also some *O.litigiosa*) and the pennant diatoms *Pseudonitzschia subcurvata* and *Pseudonitzschia turgidulodies* (Figure 4a, b and c respectively).

The abundance of two diatom taxa was unrelated to $f\text{CO}_2$ treatment. *Chaetoceros* and *Proboscia truncata* both showed no $f\text{CO}_2$ related trend in their abundances despite being relatively large diatoms (*P. truncata* can exceed 100 µm in length) (Figure 5). This may reflect a lack of precision in the microscopy counts due to high variance among replicate fields of views. The

response of *Chaetoceros* (mainly *C.castracanei* but *C.tortissimus* and *C.bulbosus* were also present) is difficult to interpret. At day 14 their abundance was lower in the three higher $f\text{CO}_2$ treatments. By day 18 no $f\text{CO}_2$ related trend was evident in *Chaetoceros* abundance (Figure 5b).

### 3.2.2 Flagellates

Colonial life stage *Phaeocystis antarctica* occurred in much higher abundances in the three lower $f\text{CO}_2$ treatments. It was

the most abundant flagellate in this study, reaching concentrations from $\sim 1.0\text{x}10^5$ to $1.26\text{x}10^7 \text{cells L}^{-1}$ at $f\text{CO}_2$ levels $\leq$634 µatm (Figure 6). This starkly contrasted with abundance at $f\text{CO}_2$ levels $\geq$953 µatm in which they did not exceed $1.6\text{x}10^6 \text{cells L}^{-1}$.

Other flagellate taxa in our study showed a variety of responses. The choanoflagellate *Bicosta antennigera* showed a $f\text{CO}_2$ related response similar to that of *P.antarctica* with higher abundances in $f\text{CO}_2$ treatments $\leq$506 µatm and lower abundances




at higher $f\mathrm{CO}_2$ levels (Figure 7a). Other choanoflagellates (mainly *Diaphanoeca multiannulata*) showed no consistent trend in response to $f\mathrm{CO}_2$ level (Figure 7b).

The abundances of other nanoflagellates and other heterotrophic protists were low and seemingly unrelated to $f\mathrm{CO}_2$ treatment. All other nanoflagellates were low in abundance and had high variance between field of view counts, therefore these were grouped together. This group, "Other Flagellates", including species *P.antarctica* (gamete and flagellate forms), *Telonema antarctica*, *Leucocryptos* sp., *Polytoma* sp., *Pyramimonas gelidicola*, *Geminigera* sp., *Mantoniella* sp., *Bodo* sp., *Triparma laevis* subsp *ramisping*, *T.laevis* subsp. *pinnatilobata* and an unidentified haptophyte. Similarly microheterotrophs comprised only of ~1% of all cells and was dominated by an unidentified euglenoid (~0.8%). Dinoflagellates were grouped into autotrophic dinoflagellates (mainly *Gymnodimium* and *Heterocapsa*) and heterotrophic dinoflagellates (predominantly *Gyrodinium* sp., *G. glaciale*, *G. lachryma* and *Protoperidinum* cf. *antarcticum*). Ciliates were grouped together but were mostly comprised of *Strombidium* sp. While none of these functional groups showed a trend in abundance correlated with $f\mathrm{CO}_2$ treatment, this may be due to the abundance of these taxa being poorly resolved.

### 3.3 Community-level responses

Analysis of the community-level responses has been separated into the 8 day acclimation and 10 day growth periods. During acclimation the growth of the cells was limited by low light. SIMPROF analysis of the microbial community over the acclimation period identified three significant groups (p <0.05) (Figure 8). Group 1 is comprised of all the treatments at day 1, group 3 contains of all the treatments over days 3, 5 and 8 except for the lowest $f\mathrm{CO}_2$ treatment on day 3 (D3T1, group 2).

Cluster analysis and SIMPROF based on the composition of the protistan community identified ten significantly different groups of samples (p-value <0.05) during the growth period (days 8 to 18) (Figure 9a). On days 8 and 10 the communities did not differ among treatments, except for the highest $f\mathrm{CO}_2$ treatment on day 10 (D10T6), which was clustered with the day 8 samples (clusters 8 and 9, Figure 9a). Day 12 treatments are scatter across the cluster groups but day 14 samples are again grouped together (all treatments except 634 µatm together in cluster 6). On day 16 the $f\mathrm{CO}_2$ treatments were clustered together except at the highest $f\mathrm{CO}_2$ level. By day 18 the $f\mathrm{CO}_2$ treatments had separated into two distinctly different groups; one with the three lowest $f\mathrm{CO}_2$ treatments and the second with the three highest $f\mathrm{CO}_2$ treatments. Interestingly, these three highest $f\mathrm{CO}_2$ treatments fall into the cluster with the day 16 samples (or nearby cluster 2). This shows that at day 18 the higher $f\mathrm{CO}_2$ treatments ($\geq$ 953 µatm) contained a protistan community more similar to that of day 16, and were significantly different to that of the day 18 lower $f\mathrm{CO}_2$ treatments (Figure 9a).

A nMDS in two dimensions proved a reasonable approximation to the full multivariate structure (stress = 0.05), and shows the similarity among clusters along with the relative contribution of each specific taxon/functional groups to each cluster (the more closely species are located to a sample in the nMDS, the more abundant it is in that sample) (Figure 9b). The community at day 8 was dominated by discoid centrics diatoms below 40 µm, flagellates, ciliates, *P.subcurvata* and dinoflagellates (heterotrophic dinoflagellates are located off to the left of the plot in Figure 9b). By day 18 the community had shifted to be dominated by *Fragilariopsis*, *T. antarctica*, *T. ritscheri*, *Odontella* and *P. antarctica* (Figure 9b). Between the start and the end of the experiment other taxa emerged between days 10 and 14 including an unidentified euglenoid, *L. annulata*, *B. antennigera*




and other centric diatoms. Other taxa increased in abundance between days 14 and 16, including other choanoflagellates, large centric diatoms (ie. *P. truncata* and *S. microtrias*), and pennate diatoms (ie. *P. turgidulodies*) (Figure 9b). At the end of the experiment (day 18) large diatoms, *T. ritscheri*, *Odontella* and *Fragilariopsis* (>20 μm) as well as *P.antarctica* are located close to the lower $f$CO$_2$ treatments for day 18 (cluster 1), and resistant *Fragilariopsis* (<20 μm) is found near the high $f$CO$_2$ treat-

ments (cluster 4). Interestingly, *T. antarctica* is also located close to the lower $f$CO$_2$ treatments despite being quite resistant to increased $f$CO$_2$ when analysed as a single species (Figure 9b and Figure 2b).

From the nMDS the lag in community growth and succession in the highest $f$CO$_2$ level (tank 6) can be seen. At day 10, the community is grouped with that on day 8, likewise at day 12, 14, 16 and 18 it is consistently closer to samples from the previous time point (Figure 9b). These results are consistent with the changes in total cell abundance (Figure 1). The highest

$f$CO$_2$ level inhibits growth and succession in the protistan community such that it is consistently a time point behind the other $f$CO$_2$ treatments.

CAP analysis showed differences in the trajectory of the protistan community succession over time in the different $f$CO$_2$ treatments (Figure 10). This analysis, using covariates $f$CO$_2$, NO$_x$, Si and P, provided a model which explained 71.61% of the variation in similarity among samples ($F_{4,31}$ = 19.544, p <0.001 based on 999 permutations). However, NO$_x$ was not

significant (F$_{1,32}$ = 1.3714, p >0.200 based on 999 permutations) and so was dropped from the model. In this reduced CAP model, CAP1 and CAP2 were both significant (p <0.05) (Table 2), and the remaining terms $f$CO$_2$, Si and P together accounted for 70.35% of the total variance (F$_{3,31}$ = 25.308, p <0.001 based on 999 permutations). Considering each term marginal to all others (ie. the contribution of a term after first accounting for all other terms), $f$CO$_2$ accounted for 2.92%, P 22.68% and Si 5.21% of the variance (Table 3a). The remaining terms in the reduced model were all significant when sequentially added, but

the marginal effects showed that only P and Si were significant, not $f$CO$_2$ (p >0.100) (Table 3b). The CAP analysis shows there is a separation of the protistan community between low (≤ 506 μatm), medium (634 to 1140 μatm) and high (1641 μatm) $f$CO$_2$ treatments (Figure 10). At day 18 there are two distinct treatment groups, those namely exposed to low and moderate $f$CO$_2$ (343, 506 and 634 μatm) and those exposed to high $f$CO$_2$ (953, 1140 and 1641 μatm).

## 4  Discussion

### 4.1  Acclimation to high $f$CO$_2$

Changes in the response of the protistan composition and abundance during the acclimation period of the experiment (days 1 to 8) were likely due to the transfer and establishment of the natural communities in the minicosm tanks rather than exposure to $f$CO$_2$. There was a significant shift in species composition between day one and all other samples times during acclimation. The collection of seawater (via Bambi bucket under a helicopter) and subsequent filling of the minicosms is likely to have

damaged delicate protists (Estrada and Berdalet, 1997), and the minicosm conditions may have been sub-optimal for some species (Kim et al., 2008). Therefore it is likely this change in community composition reflected the decline in delicate species.

The total abundance of the protistan community showed an initial lag in growth at the highest $f$CO$_2$ level. Our results indicate that the protists required more than 8 days to acclimate to this high $f$CO$_2$ and therefore growth was delayed compared





to other treatments. Deppeler et al. (submitted) showed that during acclimation there is a decrease in photosynthetic health of the community, but whilst all treatments had recovered by day 12, the highest $f\mathrm{CO_2}$ treatment had a greater decline in photosynthetic health and took longer to recover. This lag was also seen in accumulation of chlorophyll *a*, rate of productivity and nutrient concentrations where the slower growth meant this was the only treatment in which nutrients were not limiting by

the end of the experiment (Deppeler et al., submitted). After this lag the rate of growth in the highest $f\mathrm{CO_2}$ treatment is similar to other treatments. To our knowledge this is the first study to report such an acclimation response in a short-term experiment.

### 4.2   Autotrophic protist taxa specific responses

The microbial community within this study was highly diverse, and the detailed taxonomic classification employed allowed the range of responses by the individual taxa to be resolved. In diatoms the response was mainly size-related. Small diatoms showed

a strong resistance to high levels of $f\mathrm{CO_2}$ but large diatoms showed a non-linear response with an increase in abundance at moderate $f\mathrm{CO_2}$ levels (634-953 µatm) but a decrease a higher $f\mathrm{CO_2}$. This trend is even evident within *Fragilariopsis* cells >20 µm which showed a similar response to other large centric diatoms, but those ≤20 µm showed a higher resistance similar to other smaller diatoms. Whilst the response for most diatom taxa was related to cell size, a couple of species did not follow this trend. *Proboscia truncata* is a large cell but did not show a $f\mathrm{CO_2}$ response, likewise *Chaetoceros* did not show a response

to $f\mathrm{CO_2}$ but instead reflected the nutrient availability (see Deppeler et al. submitted for nutrient data). A number of Antarctic studies have shown an enhancement of larger diatom species with increases in $\mathrm{CO_2}$ (ie. Engel et al. 2008, Tortell et al. 2008 and Feng et al. 2010). There are also other previous studies to which our results do not align. Feng et al. (2009) found an increase in abundance of *Pseudonitzschia subcurvata* at high $f\mathrm{CO_2}$. In this study were *Pseudonitzschia* had a very low tolerance to $f\mathrm{CO_2}$ levels above the control, different to that of Feng et al. (2009).

Unlike diatom species, the dominant autotrophic nanoplankton, *Phaeocystis antarctica*, dramatically declined in abundance at the three highest $f\mathrm{CO_2}$ levels. There has been a common consensus in other ocean acidification studies that pico- and nanoplankton abundance increases at high $\mathrm{CO_2}$ levels (ie. Hare et al. 2007, Brussaard et al. 2013, Davidson et al. 2016, Thomson et al. 2016 and a recent review by Schulz et al. 2017), but our study only finds this responses in diatoms. *P.antarctica* had an increase in abundance with moderate $f\mathrm{CO_2}$ levels but had a strong threshold level between 634 and 963 µatm, above which

abundances were very low. This contrasts with the findings of many ocean acidification studies that found that *P.antarctica* abundance was either not effected or increased by elevated $\mathrm{CO_2}$ at the experimental $\mathrm{CO_2}$ levels used in their studies (Tortell et al., 2008; Feng et al., 2010; Trimborn et al., 2013). It is noted that these studies were conducted in the Ross Sea which is a very different ecosystem to other coastal Antarctic areas (Smith et al., 2014; Deppeler and Davidson, 2017). The difference in response of *P.antarctica* compared to other studies showing an increase in pico- and nanoplankton (Schulz et al., 2017), may

be because of the peculiar physiology of this alga in the colonial life stage (which dominated in our study). *P.antarctica*, when in colonial form, protect their cells from intimate contact with the surrounding seawater by a thin tough skin and behave in a similar fashion to a large diatom (Davidson and Marchant, 1992; Hamm et al., 1999; Hamm, 2000).

This study's $f\mathrm{CO_2}$ treatments extend well past the range of other studies, where commonly the highest level was between 750 and 1000 µatm compared to 1641 µatm in this study. When the results of this study are compared in light of these $\mathrm{CO_2}$





levels the conclusions align. For example the increase in large diatoms found by Engel et al. (2008), Tortell et al. (2008) and Feng et al. (2010) was at $CO_2$ levels between 700 and 800 µatm, agreeing with the increase in large diatoms in this study at $fCO_2$ between 343-634 µatm. Again an increase in *P.antarctica* reported by Tortell et al. (2008), Feng et al. (2010) and Trimborn et al. (2013) was to $CO_2$ levels between 750 and 800 µatm, again agreeing with the increase in *P.antarctica*

abundance in this study. Xu et al. (2014) observed a non-linear response by *P.antarctica*, with an increase in abundance at 600 µatm $CO_2$ but a decrease at 800 µatm, similar to the finding of this study. This non-linear response could explain the variation in other microbial ocean acidification studies. Previous studies have shown a range of responses to $CO_2$, however it is possible that their highest $CO_2$ level is sitting either just below or above the tipping point for their study species or community.

It has been hypothesized that phytoplankton will benefit from increased $CO_2$ due to RuBisCo's low affinity for $CO_2$ (the

carbon fixation enzyme in photosynthesis) (Reinfelder, 2011). The half saturation constant of RuBisCo for $CO_2$ is substantially higher than occurs in the ocean, and it has been proposed that the anthropogenic rise in ocean $CO_2$ may enhance the rates of phytoplankton photosynthesis (Rost et al., 2008). Most phytoplankton species have highly regulated carbon concentrating mechanisms (CCMs) which enhances the $CO_2$ available for photosynthesis by increasing the $CO_2$ supply so that it is less rate-limiting (Reinfelder, 2011). Yet the beneficial effects of enhanced $CO_2$ availability are offset against the coincident increase

of $H^+$ ion concentration. It is thought that the energy saved by decreased CCM activity will be outweighed by the energy required to increase intracellular processes that mitigate this $H^+$ increase within the cell (ie. the proton pump) (Taylor et al., 2012). Our findings suggest a combination of these two effects. The increase in abundance with moderate increases in $fCO_2$ evident for many species could be due to the increased availability of $CO_2$ and down-regulation of CCMs, therefore allowing energy saved from CCM activity to be used in other cellular processes. However there is a limit to this, a further decrease in

pH, and increase in $H^+$ ions, results in an increase in energy used to maintain the homeostasis of the cell pH. Deppeler et al. (submitted) reports the effects of $fCO_2$ on photosynthetic physiology during our study, showing that CCM activity has been down-regulated in the highest $fCO_2$ treatment of the experiment but not in at the lowest $fCO_2$. This supports the theory that whilst the CCM activity might be down-regulated, the community is still being inhibited at high $fCO_2$ level by the metabolic cost of $fCO_2$ tolerance mechanisms such as proton pumps.

**4.3 Response of heterotrophic protists**

Intriguingly, we found different $CO_2$-induced responses by the two dominant choanoflagellate taxa, *Bicosta antennigera* and other choanoflagellates (>90% consisted of *Diaphanoeca multiannulata*). To our knowledge this is the first study to show differing responses of choanoflagellate species abundances due to ocean acidification. Previous studies have reported no effect (Moustaka-Gouni et al., 2016a, b) or a decrease (Davidson et al., 2016) of choanoflagellates to increased $CO_2$, but this is the

first to show a different responses amongst taxa. The reasons for choanoflagellate taxa responding differently to elevated $CO_2$ are unclear. Gong et al. (2010) showed that lorica formation in choanoflagellates can be effected by pH changes but SEM preparations in this study found no evidence of this. Some studies of sperm flagella have suggested that increased $CO_2$ can slow metabolic rates or interrupt flagella function (Havenhand and Schlegel, 2009; Morita et al., 2009). Thus difference in sensitivity to $CO_2$ may reflect the differences in lorica complexity, cellular morphology or physiology among taxa.





If the species-specific response of choanoflagellates is indicative of the behaviour of the broader microheterotrophic community, this finding raises a previously unseen level of complexity into the effect of ocean acidification on microbial communities. Previous studies have observed no direct $CO_2$ related response on microheterotrophic protozoa community composition (Suffrian et al., 2008; Aberle et al., 2013). Unfortunately the abundance of protistan heterotrophs other than choanoflagellates

was generally low with high variability, making it difficult to detect any $CO_2$ response. If the species-specific responses of choanoflagellates seen in this study are indicative of $CO_2$-induced responses by other microheterotrophic grazers (e.g. autotrophic dinoflagellates and ciliates), the implications for top-down control of protists and prokaryotes could be quite profound.

## 4.4   Community-level responses

This study shows that there is a significant shift in the protistan community structure with increasing $f\mathrm{CO}_2$. The community response is not linear with an increase in abundance between 343 to 634 µatm favoring larger centric diatoms and *Phaeocystis antarctica*, however there is a threshold, and above 634 µatm there is decrease in abundance and shift towards smaller *Fragilariopsis*. While the nMDS and CAP showed that the primary driving factor behind community change was time (sample day, or nutrient concentration as a proxy), a significant $f\mathrm{CO}_2$ induced response was observed. This non-linear response has been

previously observed in individual taxa response to increased $CO_2$ (Trimborn et al., 2013; Xu et al., 2014). Community level studies have reported a shift in community composition with increased $f\mathrm{CO}_2$ (Schulz et al. 2017 and refs therein), but what has not been reported before is the non-linear response at the community level such that has observed in our study.

Comparisons with Davidson et al. (2016) and Thomson et al. (2016) show that there is a consistent threshold $f\mathrm{CO}_2$ above which protistan communities at this site alter their structure. A similar series of minicosm experiments was conducted in the

same location, Prydz Bay East Antarctica, in the austral summer of 2008-09 (Davidson et al., 2016; Thomson et al., 2016). That study found a significant shift in the microbial community above $f\mathrm{CO}_2$ levels between 750 to 1281 µatm. As here, once that threshold had been reached there was a shift in the community structure towards smaller cells and a decrease in protistan abundance. Thomson et al. (2016) reports results from three different starting communities and nutrients levels, but despite these differences the threshold remained the same. Our study differs from that of Davidson et al. (2016) and Thomson

et al. (2016) in that it has a narrower $f\mathrm{CO}_2$ range in the treatments, and also included an acclimation period. Despite these differences, the threshold level found in our study falls within that of Davidson et al. (2016) and Thomson et al. (2016). Unlike those previous studies, we saw an increase in large protists below this threshold level. This could be due to having six treatments across a smaller $f\mathrm{CO}_2$ range, therefore allowing a higher resolution in the response of the community prior to the threshold. It could also be due to the inclusion of the acclimation period, giving cells time to adjust to the $f\mathrm{CO}_2$ level prior to the beginning

of growth and therefore allowing them to capitalise on the benefits of moderate $f\mathrm{CO}_2$. The results from these studies show that there is a consistent $f\mathrm{CO}_2$ threshold that elicits changes in the structure of microbial communities in near-shore waters of Prydz Bay, East Antarctica, both within a season and among seasons. Furthermore, irrespective of including an acclimation phase, the nature of the change in the protistan communities at high $f\mathrm{CO}_2$ remains similar, though the magnitude can change greatly.



The flow on effect of decreased abundance and a structural shift in the protistan community to smaller cells could have far-reaching impacts through effects on the rest of the near-shore Antarctic food-web and biogeochemical cycles. Many studies have shown that a shift in protistan community composition can effect the palatability, nutritional quality and availability of phytoplankton cells available to grazers and the higher trophic levels (Rossoll et al., 2012; Caron and Hutchins, 2013; Bermúdez et al., 2016; Davidson et al., 2016). Antarctic microbes are also a vital component of many elemental cycles, ocean biogeochemistry and provide important roles in the feedback of the Southern Ocean to global climate change (Arrigo and Thomas, 2004; Arrigo et al., 2008; Kirchman, 2008). The results of this study show that the abundance of *Phaeocystis antarctica* could significantly change with future increases in $f\text{CO}_2$. This species is particularly important in a number of near-shore Antarctic nutrient cycles through their substantial production of dimethyl sulfide, which acts as a cloud condensation nuclei when released into the atmosphere (Liss et al., 1994). *P.antarctica* also plays a vital role in the carbon flux when in colonial form. Davidson and Marchant (1992) show that the majority of *Phaeocystis* biomass remains unutilized and therefore enters the carbon flux as dissolved organic carbon. Likewise, larger diatom species are also important in the sequestration of carbon to the deep ocean through their role in the vertical carbon flux (Passow and Carlson, 2012; Caron and Hutchins, 2013). The effect of a protistan community dominated by smaller cells on this vertical flux is uncertain, but any decline in this flux would have a positive feedback on atmospheric $\text{CO}_2$ level, as instead of being sequestered to the deep ocean it would be respired in the near-surface waters and released into the atmosphere.

## 5 Conclusions

Returning to the aims of this study, there are four main conclusions;

1. The responses to increased $f\text{CO}_2$ was taxon-specific in both autotrophic and heterotrophic protists, with different taxa showing different tolerance thresholds for $f\text{CO}_2$. In diatoms this response is mainly driven by cell size, with smaller cells showing a high tolerance to increased $f\text{CO}_2$ and larger cells a lower tolerance. This trend is consistent even within a taxon as demonstrated by the large size ranging *Fragilariopsis* cells in this experiment. Whilst size-related responses have previously been observed for nano- and pico plankton vs larger cells (see Schulz et al. 2017 for a review), our study shows that this trend continues in diatoms greater than 20 µm.

2. An increase in $f\text{CO}_2$ significantly changes the composition and abundance of protists in this coastal East Antarctic community. The threshold for this response is between 634 and 953 µatm. Below this threshold there is an increase in community abundance and the community was characterised by large centric diatoms and *Phaeocystis antarctica*. Above the threshold there was a decrease in abundance and the community was dominated by smaller diatoms predominantly *Fragilariopsis* below 20 µm.

3. During the eight day acclimation period there were no differences in community structure between $f\text{CO}_2$ treatments, however a lag in protistan growth until day 12 of the experiment was observed at 1641 µatm. This suggests that the





acclimation period was not sufficient for the community to adjust its composition and/or cellular physiology to cope with this high $f\mathrm{CO_2}$ level.

4. Comparisons with Davidson et al. (2016) and Thomson et al. (2016) show that this threshold level is not only consistent across a season but also between years for protistan communities at Prydz Bay, East Antarctica.

The results of this study suggest that there is a strong threshold level above which the structure of this near-shore Antarctic microbial community significantly changes, and this threshold is around the $\mathrm{CO_2}$ level predicted for the end of this century (IPCC, 2014). This change could have significant flow-on effects to the coastal Antarctic ecosystem as it would threaten the many ecosystem services that marine microbes provide, and result in cascading effects through the Antarctic food-web, elemental cycles, ocean biogeochemistry and feedbacks on global climate change.

*Code and data availability.*   Data is available via the Australian Antarctic Division Data Centre:

Hancock, A.M., Davidson, A.T., McKinlay, J., McMinn, A., Schulz, K., van den Enden, D. (2017, updated 2017) Ocean acidification changes the structure of an Antarctic coastal protistan community Australian Antarctic Data Centre - doi:10.4225/15/592b83a5c7506.

Code is available via the Australian Antarctic Division Data Centre:

Hancock, A.M., Davidson, A.T., McKinlay, J., McMinn, A., Schulz, K., van den Enden, D. (2017, updated 2017) Ocean acidification

changes the structure of an Antarctic coastal protistan community Australian Antarctic Data Centre - CAASM Metadata (https://data.aad. gov.au/metadata/records/AAS_4026_Microscopy_Multivariate_Statistics_Rcode)

*Author contributions.*   A. Davidson designed the research and led the minicosm experiment at Davis Station, Antarctica and all carbonate chemistry measurements, calculations and manipulations were performed by K. Schulz. A. Davidson collected the samples during the experiment and A. Hancock performed all light and electron microscopy work with assistance from A. Davidson and R. van den Enden. A.

Hancock conducted the data and statistics analysis using R code and statistical approach developed by J. McKinlay. A. Davidson and J. McKinlay provided assistance and guidance with data and statistical analysis and interpretation. A.Hancock prepared the manuscript with contributions from all co-authors.

*Competing interests.*   The authors declare no competing interests.

*Acknowledgements.*   This work was supported by the Australian Government Department of Environment and Energy as part of Australian

Antarctic Science project 4026 at the Australian Antarctic Division, as well as Australian Research Council's Special Research Initiative for Antarctic Gateway Partnership (Project ID SR140300001) and the Australian Governments Cooperative Research Centres Program through the Antarctic Climate and Ecosystems Cooperative Research Centre (ACE CRC). We would like to acknowledge the Australian Antarctic Division for technical and logistical support, as well as the other members of the minicosm experimental team at Davis Station 2014 and the



expeditioners at Davis Station the summer of 2014 and 2015 for their support and assistance. We would also like to thank Simon Reeves for providing R code that created the temporal graphs.





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





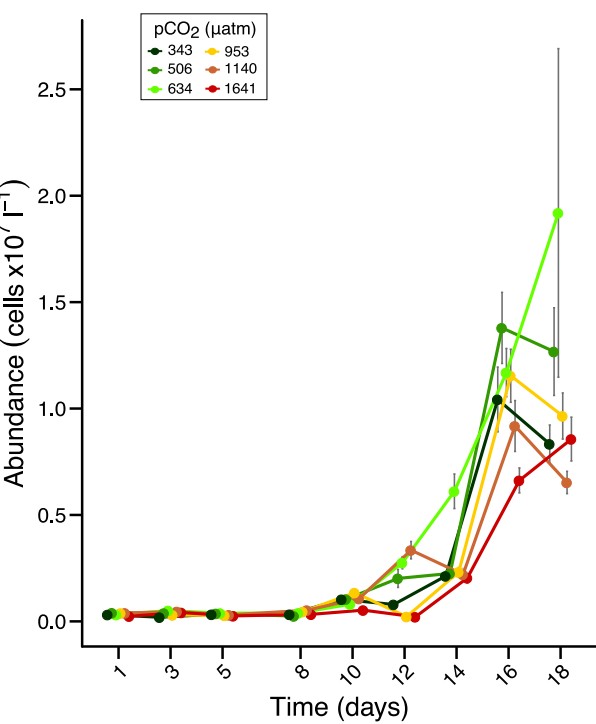

**Figure 1.** Microscope-derived abundance counts of protists over an 18 day incubation of a natural protistan community in tanks maintained at different $f\mathrm{CO}_2$ levels. Error bars are standard errors derived from pseudo-replicates undertaken at each time point for each treatment.





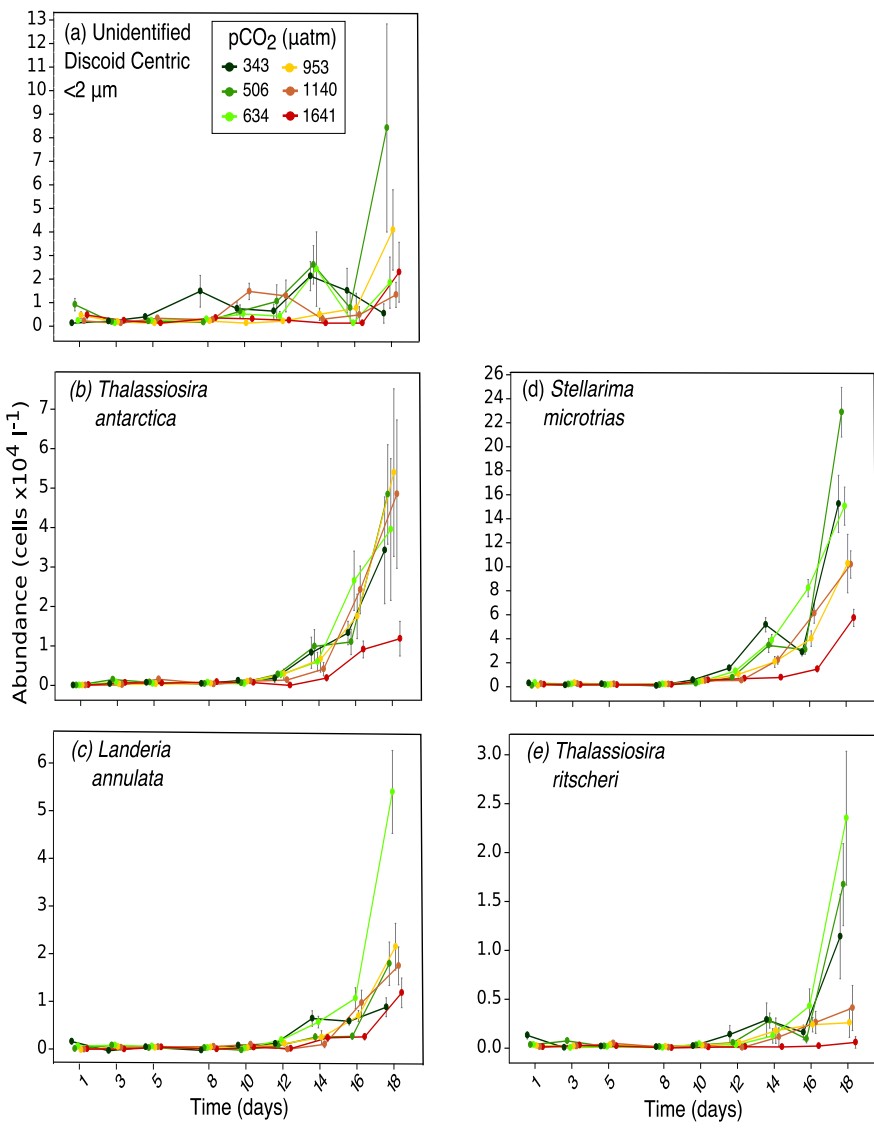

**Figure 2.** Microscope-derived abundance counts of protists. Abundances of (a) Unidentified discoid centric <2 μm, (b) *Thalassiosira antarctica*, (c) *Landeria annulata*, (d) *Stellarima microtrias*, (e) *Thalassiosira ritscheri* over days 1 to 18 of the incubation of a natural protistan community in tanks maintained at different $f\mathrm{CO_2}$ levels. Error bars are standard errors derived from pseudo-replicates undertaken at each time point for each treatment.





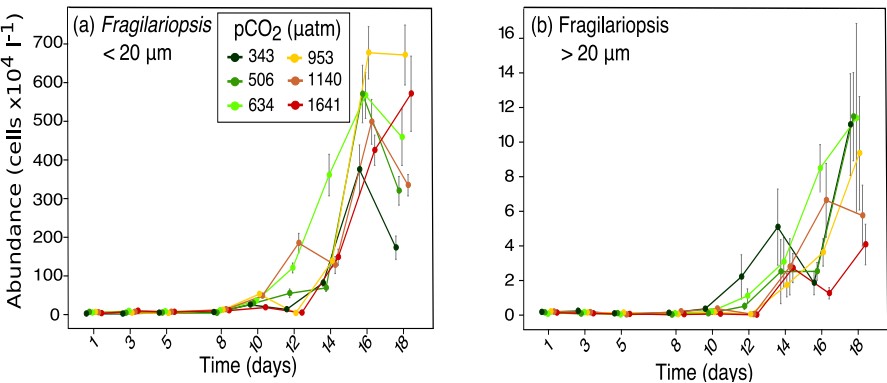

**Figure 3.** Microscope-derived abundance counts of protists. Abundances of (a) *Fragilariopsis* <20 μm, (b) *Fragilariopsis* >20 μm over days 1 to 18 of the incubation of a natural protistan community in tanks maintained at different $f\text{CO}_2$ levels. Error bars are standard errors derived from pseudo-replicates undertaken at each time point for each treatment.





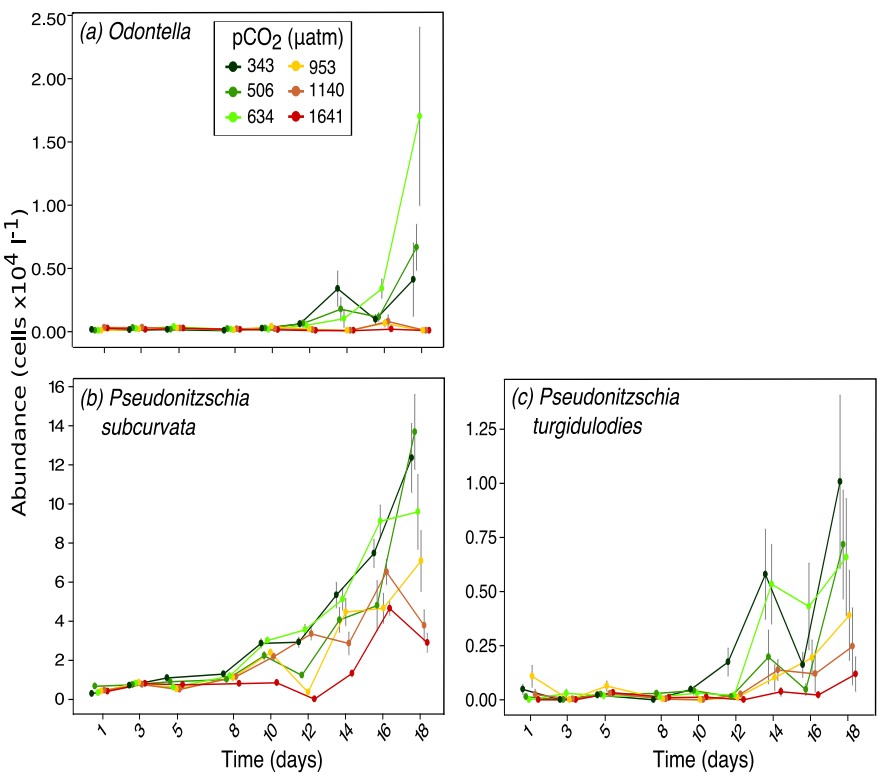

**Figure 4.** Microscope-derived abundance counts of protists. Abundances of (a) *Odontella*, (b) *Pseudonitzschia subcurvata*, (c) *Pseudonitzschia turgidulodies* over days 1 to 18 of the incubation of a natural protistan community in tanks maintained at different $f\mathrm{CO}_2$ levels. Error bars are standard errors derived from pseudo-replicates undertaken at each time point for each treatment.



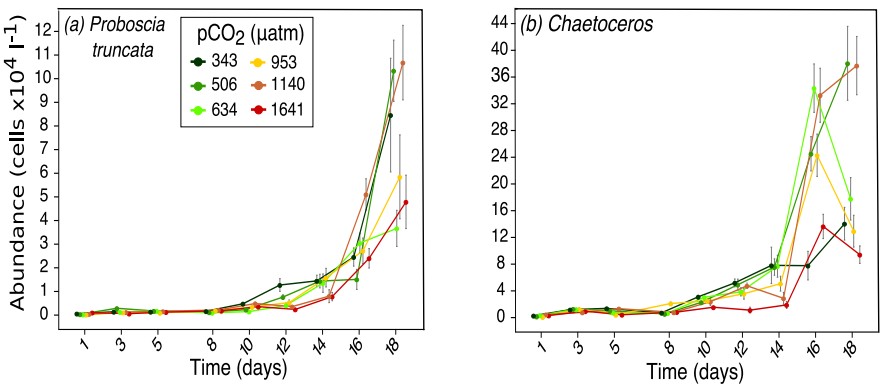

**Figure 5.** Microscope-derived abundance counts of protists. Abundances of (a) *Proboscia truncata*, (b) *Chaetoceros* over days 1 to 18 of the incubation of a natural protistan community in tanks maintained at different $f\mathrm{CO_2}$ levels. Error bars are standard errors derived from pseudo-replicates undertaken at each time point for each treatment.



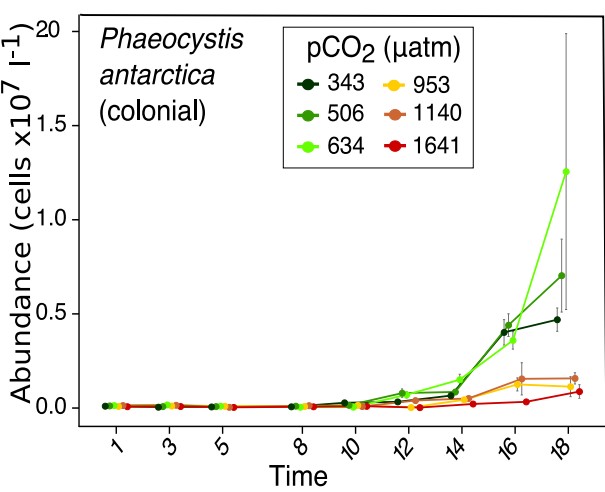

**Figure 6.** Microscope-derived abundance counts of *Phaeocystis antarctica* (colonial form) over days 1 to 18 of the incubation of a natural protistan community in tanks maintained at different $f\mathrm{CO_2}$ levels. Error bars are standard errors derived from pseudo-replicates undertaken at each time point for each treatment.





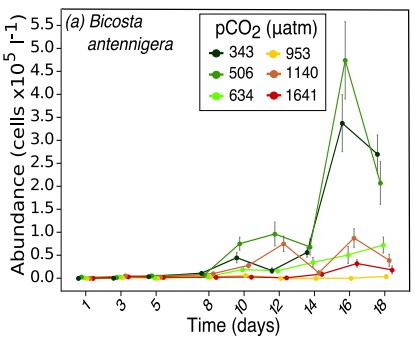
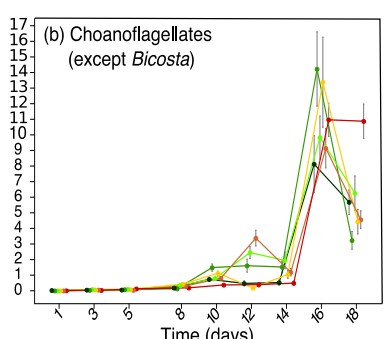

**Figure 7.** Microscope-derived abundance counts of protists. Abundances of (a) *Bicosta antennigera*, (b) Choanoflagellates (except *Bicosta*) over days 1 to 18 of the incubation of a natural protistan community in tanks maintained at different $f\mathrm{CO_2}$ levels. Error bars are standard errors derived from pseudo-replicates undertaken at each time point for each treatment.





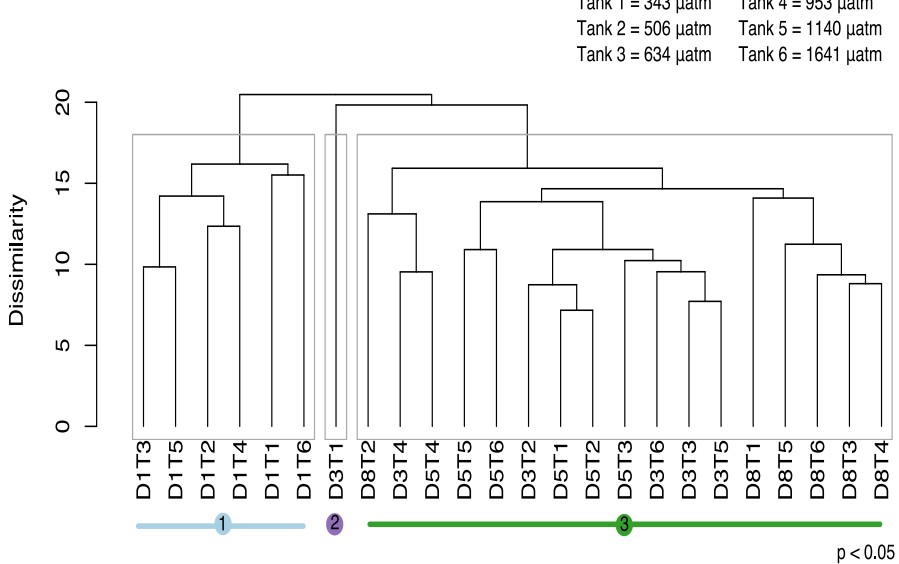

**Figure 8.** Cluster analysis based on similarity in protistan community structure among $f\mathrm{CO_2}$ treatments and times during the acclimation period (days 1 to 8). The analysis shows three significantly different groups obtained by SIMPROF (denoted by grey boxes around clusters and coloured lines beneath sample labels). Samples are abbreviated according to days of incubation (D1-8) and level of $f\mathrm{CO_2}$ treatment (T1-6 for 343, 506, 634, 953, 1140, 1641 μatm respectively).





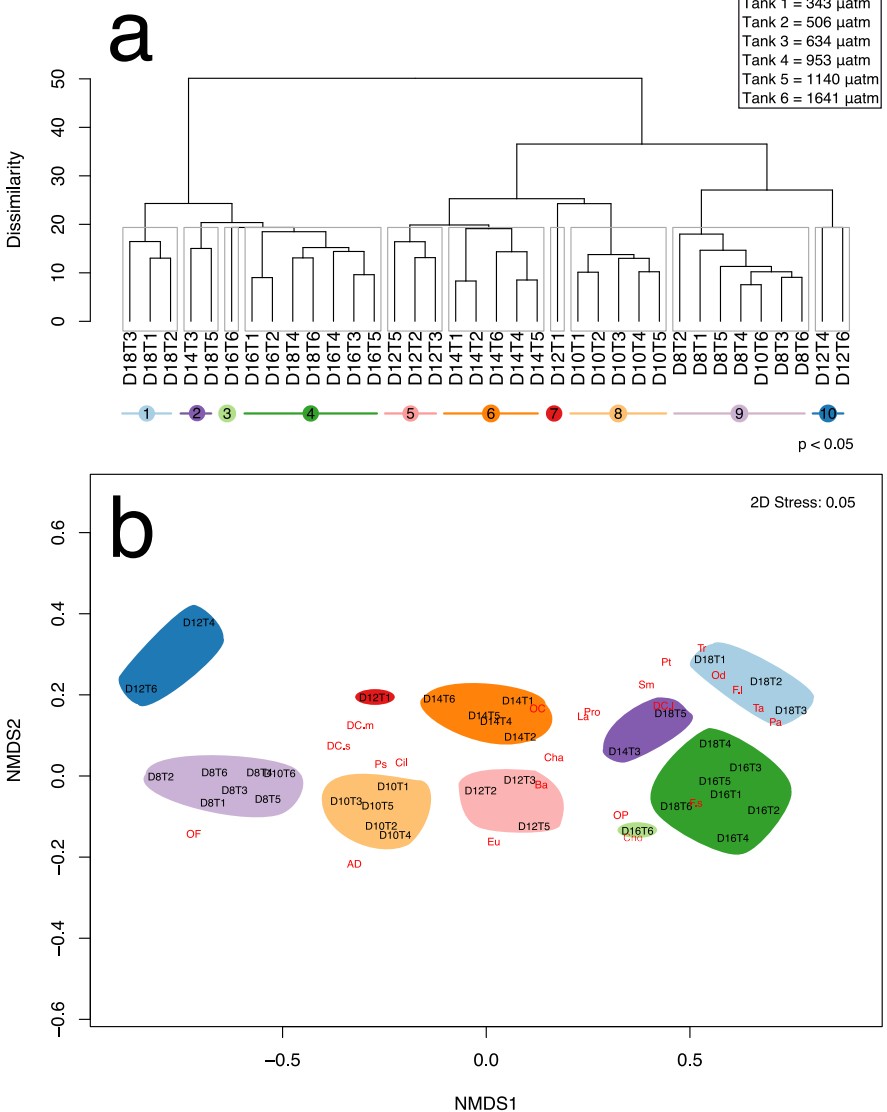

**Figure 9.** Cluster analysis and nMDS based on similarity in protistan community structure among $f\mathrm{CO_2}$ treatments and times over days 8 to 18 of the incubation. (a) The cluster analysis shows ten significantly different groups obtained by SIMPROF (denoted by grey boxes around clusters and coloured lines beneath sample labels). (b) nMDS plot structure showing the unconstrained ordination of dissimilarities in protistan community structure with time and $f\mathrm{CO_2}$ treatment in 2 dimensions, overlaid with weighted-averages of the day-treatment scores for each protistan taxa/functional group (see Table 1 for abbreviations). Samples are abbreviated according to days of incubation (D8-18) and level of $f\mathrm{CO_2}$ treatment (T1-6 for 343, 506, 634, 953, 1140, 1641 µatm respectively).



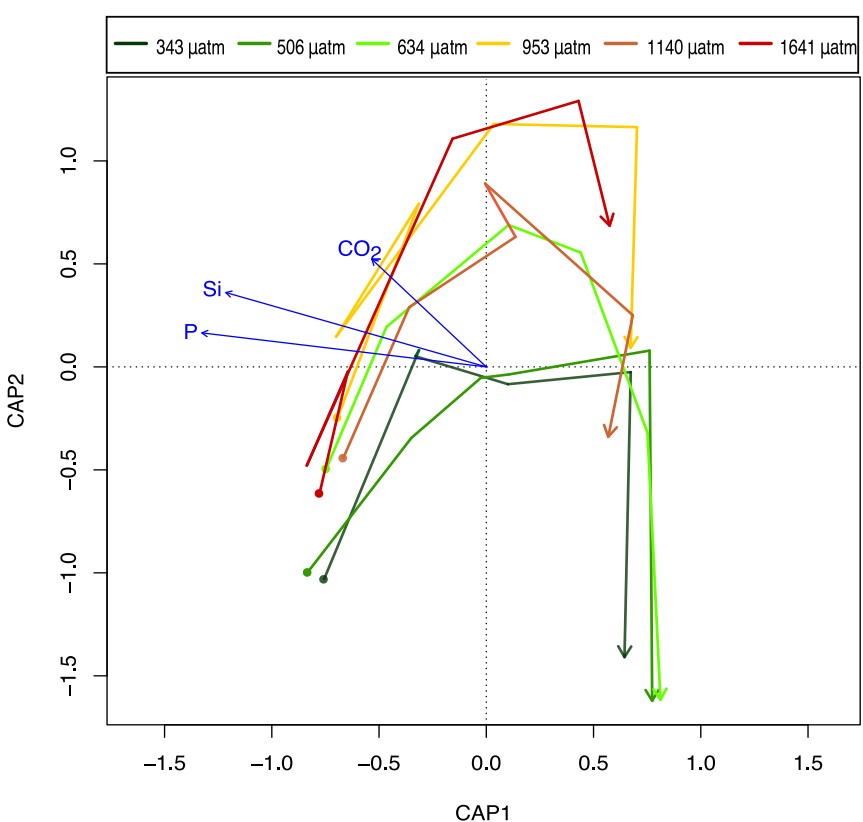

**Figure 10.** Canonical analysis of principal co-ordinates (CAP) based on the similarity in protistan community structure among $f\mathrm{CO}_2$
treatments and times over days 8 to 18 of the incubation, showing the trajectory of change in the protistan community for each $f\mathrm{CO}_2$ level
(coloured arrows) based on the abundance of the component taxa/functional group. Arrow starting points are day 8 and all arrows end on day
18 of the experiment. Linear projections of significant constraints $\mathrm{CO}_2$, Si and P appear as blue linear arrows.



**Table 1.** Protistan group abbreviations.

| Taxon\Functional Group | Abbreviation |
| --- | --- |
| Autotrophic Dinoflagellates | AD |
| *Bicosta antennigera* | Ba |
| *Chaetoceros* | Cha |
| Choanoflagellates (except *Bicosta*) | Cho |
| Ciliates | Cil |
| Discoid Centric Diatoms >40 µm | DC.l |
| Discoid Centric Diatoms 20 to 40 µm | DC.m |
| Discoid Centric Diatoms <20 µm | DC.s |
| Euglenoid | Eu |
| *Fragilariopsis* >20 µm | F.l |
| *Fragilariopsis* <20 µm | F.s |
| Heterotrophic Dinoflagellates | HD |
| *Landeria annulata* | La |
| Other Centric Diatoms | OC |
| *Odontella* | Od |
| Other Flagellates | OF |
| Other Pennate Diatoms | OP |
| *Phaeocystis antarctica* | Pa |
| *Proboscia truncata* | Pro |
| *Pseudonitzschia subcurvata* | Ps |
| *Pseudonitzschia turgidulodies* | Pt |
| *Stellarima microtrias* | Sm |
| *Thalassiosira antarctica* | Ta |
| *Thalassiosira ritscheri* | Tr |

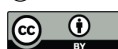



**Table 2.** Canonical analysis of principal coordinates (CAP) axis significance against covariates. Permutation tests assessing the significance of each constrained axis in CAP using the covariates $f\mathrm{CO_2}$, P and Si as constraints upon community structure.

|          | df | Variance | $F$    | No. Perm | Pr ($>F$) |
|----------|----|----------|--------|----------|-----------|
| CAP1     | 1  | 2.00156  | 634212 | 999      | 0.001     |
| CAP2     | 1  | 0.07228  | 2.2902 | 999      | 0.038     |
| CAP3     | 1  | 0.05339  | 1.6916 | 999      | 0.105     |
| Residual | 32 | 1.00992  |        |          |           |



**Table 3.** Permutation tests assessing the significance of each environmental covariate (constraint) in determining protistan community structure using principal co-ordinates (CAP), showing significance of each term when (a)sequential added (b)marginal effects for $f\text{CO}_2$, P and Si.

| (a) | df | Variance | F | No. Perm | Pr (>F) |
|---|---|---|---|---|---|
| $f\text{CO}_2$ | 1 | 0.25848 | 6.237 | 999 | 0.003 |
| P | 1 | 1.75541 | 62.7309 | 999 | 0.001 |
| Si | 1 | 0.11072 | 3.9566 | 999 | 0.03 |
| Residual | 32 | 0.89546 | | | |
| (b) | df | Variance | F | No. Perm | Pr (>F) |
| $f\text{CO}_2$ | 1 | 0.06197 | 2.2144 | 999 | 0.12 |
| P | 1 | 0.48178 | 17.2167 | 999 | 0.001 |
| Si | 1 | 0.11072 | 3.9566 | 999 | 0.027 |
| Residual | 32 | 0.89546 | | | |