# Peer review of "Ocean acidification changes the structure of an Antarctic coastal protistan community"

_Biogeosciences, 2017_

## Referee Comment (RC1) · Anonymous Referee #1 · 6 Jul 2017

General comments: This study investigates the effect of elevated $CO_2$ concentrations on prostistan community composition of Prydz Bay, East Antarctica. As the quantification of cell abundances at the species level is very labor intensive, often studies tend to neglect this very important aspect that has been considered in detail in the study by Hancock and co-authors. The data presented are interesting and I belief it should be published, but it needs a considerable revision to be acceptable for Biogeosciences. The authors need elaborate in more detail about the counting procedure, in particular for cells which were present only in low abundance as they often tend not to be evenly distributed in the Utermöhl chamber, causing easily wrong cell abundance estimates. For better readability of the manuscript, information on carbonate chemistry as well as on macronutrient concentrations over the course of the experiments is

needed. In particular, the onset of nutrient limitation on day 16 needs to be accounted for in the discussion of the development of the protistan community, which has been neglected so far. At the moment, the discussion mainly concentrates solely on the $CO_2$ effects, which is fine until day 15, but not after this time point. This aspect needs to be addressed. For better and faster comparability of the figures of species-specific cell abundances, I recommend to use the unit 'cells per mL'. Furthermore, to strengthen the author's argument that growth of large-sized diatoms is more prone to high $CO_2$ concentrations, a graph showing actually the different trends in total abundance of all small versus all large diatoms, similar to figure 3, is needed.

Introduction: P2, L5-6: This statement is not right as there are several studies that were already published on OA effects in various natural assemblages of Southern Ocean microbes (Tortell et al. 2008, Feng et al. 2010, Hoppe et al. 2013, McMinn et al. 2014, Young et al. 2015, Coad et al. 2016, Davidson et al. 2016, Thomson et al. 2016). Please rephrase. P2, L19-35: Considering that the authors already cited 8 papers that were published on $CO_2$ effects, it is not really appropriate to write that "there have been relatively few studies". Please also cite the studies by Hoppe et al. 2013 PLOS One and Young et al. 2015 MEPS, which are currently missing. The latter two studies need also to be taken into account when summarizing the findings on $CO_2$-dependent shifts in community composition in this paragraph. P2, L23-25: Please note that Feng et al. (2010) reported a shift from Cylindrotheca to Chaetoceros from 380 to 750 $\mu$atm $pCO_2$, and not from Pseudo-nitzschia. Further Tortell et al. (2008) did not observe a $CO_2$-triggered shift in Phaeocystis antarctica. It was reported that both summer and spring phytoplankton communities were dominated by P. antarctica and within the communities a shift among diatoms was observed.

Methods: P3, L17: Did the authors assess whether the gravity filtration procedure intro-duced cell damage and/or physiological fitness of the sampled microbial community? The latter could have affected the evolution of the community structure. P3, L25: To me is unclear why during the initial acclimation phase the community was exposed to

the extremely low light intensity of ∼1 $\mu$mol photons m-2 s-1. P3, L28-32: How was the light intensity adjusted? Were the minicosms not exposed to the natural irradiance cycle? Did the authors monitor daily in situ irradiances over the whole experiment? The manipulation of the light intensity remains unclear to me. P4, L2-12: I can understand that carbonate chemistry results are reported in detail in Deppeler et al. (submitted), but also for this manuscript there is the need to give information on the successful CO2 manipulation of each CO2 treatment at least in a table. For the interpretation and discussion on the results of the development of the community composition, it would be also helpful to give the information on carbonate chemistry (e.g. pH, fCO2) at the day of seawater sampling. P5, L2-11: The counting of particularly large diatoms can be problematic. To this end, it is recommended to count the whole Utermöhl chamber as species are not distributed evenly. In particular, chain-forming diatom species can be very patchy, making their quantification on the basis of 20 chosen fields of view difficult. How many cells did the authors count per species? How was the patchiness of species distribution within the chamber accounted for? Considering the low cell numbers, it is important to address this issue as otherwise easily wrong cell abundance estimates can be made.

I miss information on the development of the macronutrient (N, P, Si) concentrations over the duration of the experiment. This info needs to be provided either in a table or a figure. According to Deppeler et al. (submitted to Biogeosciences) N was depleted for most treatments at day 16, this means that in addition to the changes in fCO2 N also potentially acted as stressor at the end of the experiment, potentially influencing community composition at the end of the experiment. The latter information is not obvious when nutrient data are not shown in this manuscript and needs to be accounted for in the interpretation and discussion of the results. Hence, to assess the effect of increasing fCO2 levels on community composition, the authors should rather compare results at day 16 instead of day 18. For instance, the abundance of Fragilariopsis species < 20 $\mu$m of the 343, 506, and 634 $\mu$atm fCO2 treatments strongly dropped between day 16 and 18, coinciding with nitrate limitation at these specific fCO2 levels.

Results: P6, L21-22: It seems very unlikely that the high variation in protist abundance of the 635 $\mu$atm pCO2 treatment, accounting for $\sim$10000 cells per mL, comes from the increase in rare large cell diatom species, which were only present between 5 up to 200 cells per mL (Fig. 2-5). P6, L27-29: It is not really clear which figure underlines this statement. Also, it would be helpful to point out which diatoms were classified as 'large' and 'small'. For this statement, it would be good to have a graph showing actually the different trends in total abundance of all small versus all large diatoms, similar to figure 3. P6, L30: Do the authors now refer to Fig. 2b-d when they refer to discoid centric diatoms or is Fig. 2a meant, but then it does not make sense to refer to 'unidentified discoid centric'. Does the latter term refer to one single species or does it summarize all counts of unidentified discoid diatom species that were smaller than 2 $\mu$m? Another idea would be to add the cell volume of the species next to its name on the graph, making it easier to see the size differences at a glance. P7, L14: To what fCO2 treatment does the control refer to? 343 $\mu$atm? P7, L12-14: Looking at figure 3a, small Fragilariopsis cells of the 953 $\mu$atm reached highest cell abundances in comparison with all other fCO2 treatments at day 16 and 18. The authors, however, write "Abundances in the ÆŠCO2 treatments >953 $\mu$atm were lower but less than those in the control treatment..." How can this be? P7, L15-17: Why is a tolerance lower when cell abundance is higher? P7, L20-21: The species name is O. weissflogii and not weissfloggi. Also write 'pennate' instead of 'pennant'. Also, it is Pseudonitzschia and not Pseudonitzschia. Also change turgidulodies to turgiduloides.

Discussion: P9, L26-28: As in almost all figures cell abundances did not change between day 1 and 8, considering also that irradiance was very low, I am surprised about the statement that community composition changed. Did the authors characterize species composition of the initial community? As the information on the characterization of the initial community is missing this complicates the interpretation on subsequent species changes through the sampling. P9, L 32-33: Taking into account the very low irradiance between day 1 and 8 the cells were exposed to, it is not surprising the community showed a severe delay in growth among all treatments, a finding which

is not mentioned here. Apparently, the combination of very low irradiance and high $fCO_2$ caused even stronger delay. This is worth to be mentioned. On which observation is the statement based that 'the protists required more than 8 days to acclimate to this high $fCO_2$'? P10, L1-6: To underline the statement that community growth of the highest $fCO_2$ treatment was lowest, why do the authors not calculate community growth rate? All data are there and this would strengthen their argumentation. P10, L9-15: I am not yet convinced about the statement that in 'diatoms the response was mainly size-related'. To underline this, a graph showing actually the different trends in total abundance of all small versus all large diatoms, similar to figure 3, is required. The authors even point out that 'a couple of species did not follow this trend'. P10, L9-19: The fact that nutrients became limiting either on day 16 or 18 needs to be elaborated in more detail. This aspect was fully neglected, only in L14 it is mentioned that 'Chaetoceros did not show a response to $fCO_2$, but instead reflected the nutrient availability'. This aspect needs to be discussed also for the other species. P10, L19-20: The low tolerance to high $pCO_2$ is also found and reported in Tortell et al. (2008) and Hoppe et al. (2013). P10, L20: 'Unlike diatom species, . . . Phaeocystis dramatically declined . . . at the three highest $fCO_2$ levels'. It was, however, pointed out before that 'large diatoms showed . . .a decrease at higher $fCO_2$'. There is no controversy, please modify. P10, L23: Please specify the statement 'our study only finds this response in diatoms'. To which response is referred to? The increase in diatom abundance under high $fCO_2$? But the opposite response for diatoms was claimed before. P10, L21: I disagree that there is a 'common consensus in other ocean acidification studies that pico- and nanoplankton abundance increases at high $CO_2$ levels'. Like the dataset of the authors, there are several studies reporting the opposite for Southern Ocean communities (Tortell et al. 2008, Feng et al. 2010, Hoppe et al. 2013). Please rephrase more carefully. P10, L23-25: Repetition, please see L20. P10, L25-28: In line with the data by Hancock et al., in none of the cited studies Phaeocystis antarctica showed a positive growth response to high $CO_2$, growth rather remained unaffected by $CO_2$. Please also add Trimborn et al. 2017 Physiol. Plant, which is in line with the latter

observation. P10, L28-30: I disagree. The results from Feng et al. (2010) show no $CO_2$ effect on the colonial Phaeocystis antarctica. Furthermore, it is not clear to me why the presence of the mucus could have any effect on the $CO_2$ sensitivity of Phaeocystis. P11, L1: Please also cite Wu et al. 2014 that reported enhanced growth rates in response to high $CO_2$ in large diatoms. P11, L3-6: As mentioned before, no $CO_2$-dependent increase in Phaeocystis was reported in Tortell et al. (2008), Feng et al. (2010) or Trimborn et al. (2013). Please correct. Further Xu et al. did also not observe any $CO_2$-dependent increase in Phaeocystis from the current to the 2060 scenario, but a significant decline from 2060 to 2100. Please note that in the latter study next to $CO_2$, also temperature, light and Fe availability was changed, being therefore more difficult to compare with this data set here. P11, L13-15: Please also cite Trimborn et al. (2013) who actually investigated the CCM of Southern Ocean phytoplankton species, among them Phaeocystis antarctica. P11, L26-35: For better readability, please specify the direction of the observed responses of the different choanoflagellates, just saying 'there were differences' is not enough.

Discussion Part 4.4: In particular here, the onset in nutrient limitation at day 16 and 18 needs to be accounted for in the discussion of community-level responses as $CO_2$ was not the only driver. The latter statement also applies for the overall discussion.

Figures: Fig. 1-7: For better and faster comparability between cell abundances of the different species, I would use the unit 'cells per mL' in all figures instead of using 'cells x 104 L-1' as in Fig. 2-4, 'cells x 105 L-1' as in Fig. 7 or 'cells x 107 L-1' as in Fig. 1 and 6. The latter makes it even more complicated as the Y-axis is also changing, hampering a fast comparison between cell numbers between graphs of different figures. In the legends of Fig. 1 to 7, it is referred to the $pCO_2$ while in the M&M section it is referred to $fCO_2$, please stick to one of them throughout the manuscript.

---

## Referee Comment (RC2) · Anonymous Referee #2 · 15 Jul 2017

This manuscript analyses the effects of elevated CO2 on the protistan community in East Antarctica. Firstly it is great to read another biological ocean acidification study being conducted in the Antarctic as well as being a community response study. Both these areas of research are not common with many questions left unanswered. It is, therefore, particularly interesting that this study by Hancock et al. addresses community level responses in the Antarctic where biota are considered to be the most vulnerable to OA due to the rising solubility of CO2 in cold-waters. Overall this manuscript is well written, contains plentiful relevant data and attempts to close the gaps in our knowledge of important questions outstanding in the OA field. Before consideration for publication, there are a few points that need addressing. In particular, more explanation is needed about the carbonate chemistry analysis. DIC and pHT are the

[Figure]

CO2 parameters directly measured so why is fugacity of CO2 used as the CO2 parameter altered? pH (on any scale) or DIC are the usual parameters directly altered in OA studies and therefore using fCO2 limits the continuity between this study and others. I suggest using either the measured pH or DIC measurements instead. A table of differences between the carbonate chemistry of each treatment is also necessary rather than quoting Depper et al. (submitted). Many more details are generally needed. For example, how often was DIC measured? What was the variability between measurements in DIC and pH? How often was the probe calibrated? A common problem with OA research is replication. I query why this experiement was not replicated given the short duration? In addition OA research is also moving towards long-term studies spanning many months to years. I also query why such a short duration was chosen for this experiment? Throughout the manuscript there are several references to look at Deppler et al. (submitted) for information not detailed in this manuscript. I query whether this manuscript is a "stand-alone" story. Technical corrections: Page 1 line 1: remove 'of' Page 1 line 8: should be a semicolon instead of a colon. Page 1 line 8/9: is it a case of large cells decrease in abundance in high fCO2? That would be a better way to report these results as high fCO2 is the environmental stress concerned. Page 1 line 12/13: This statement needs clarification as it implies this research is not original. Page 2 line 4-6: generally OA studies on organisms higher up the foodweb in the Antarctic are few which adds importance to your study and should be mentioned with some key Antarctic papers referenced. Page 2 line 23: insert a comma. . .'With increased COÂň2, Tortell et al. . ..' Page 4 line 5: remove 'the' before adding Page 6 line 22: remove 'a' should be 'likely due to' Page 7 line 3: change to 'had increased to' Results section: removed 'show' and other variations using this word as it is unnecessary. It reads better to just say 'increased' instead of 'showed an increase'. Page 10 line 11: typo 'a' should be 'at' Page 10 line 17-19: why might there be differences between the results in this study and that of Feng et al. (2009)? Page 10 line 22: it is difficult to compare the results in this study to others quoted in this statement when different CO2 parameters were altered. Page 10 line 23: 'response' instead of 'responses'

---

## Author Comment (AC1) · 31 Aug 2017

[bg, manuscript]copernicus

We would like to thank anonymous referee #1 and #2 for their thorough and constructive review of our manuscript "Ocean acidification changes the structure of an Antarctic coastal protistan community". We are thankful to the referees for the recognition of the strengths of this paper studying the biological effects of ocean acidification on a natural community in Antarctica, an area that is rarely studied. We accept and agree with all comments; the larger changes are listed below, followed by a point per point response' to each of the referees comments.

**Introduction**

[Figure]

1. On reflection, we agree with the referees that the comparisons made to previous studies in both the introduction and discussion could be improved, and these have now been changed to more appropriately reflect the findings of previous studies and how these differ or agree with our findings.

**Methods**

1. The methods section has been expanded to include introductory paragraphs explaining the experimental design, followed by a section addressing the minicosm operation with further details on the light adjustments and intensities.

2. The text describing carbonate chemistry manipulation, sampling, measurements and calculations have been expanded, and an additional section has been included on macronutrient sampling and measurements. The data for both is presented in the supplement.

3. Additional detail has been included in the light microscopy section to address concerns of anonymous referee #1, including the additional steps taken to ensure accurate estimates of cell abundance, particularly for rare and/or species with patchy distributions.

**Results**

1. We have incorporated a discussion of taxa abundances and the community structure at time points prior to day 18 to account for the nutrient depletion towards the end of the experiment.

2. The section on size-related responses in diatoms has been amended to be clearer to the reader, and an additional figure showing responses of large (>20 μm) vs. small (<20 μm) diatoms to $f\mathrm{CO}_2$ has been included.

**Discussion**

1. The "lag in growth" discussed in the acclimation section has been removed, which on further investigation was not significant when growth rates were calculated.

2. As outlined above, the paragraphs comparing the responses of previous studies in the "Autotrophic protist taxa specific responses" section was poorly worded or at times incorrect. This has now been amended to accurately reflect the findings of previous papers and how they compare to ours.

3. A specific paragraph has been added into the discussion as part of the "Autotrophic protist taxa specific responses" considering the effect of nutrient depletion on microbial abundances.

**Figures**

1. Addition of figure showing the response of small vs. large diatoms to $f\mathrm{CO}_2$

2. $p\mathrm{CO}_2$ to $f\mathrm{CO}_2$ in figure legends

3. Amended axis to "cells $\mathrm{mL}^{-1}$" with no exponents rather than "cells x exponent $\mathrm{L}^{-1}$"

4. Average longest dimension of the cell (i.e. valve diameter or pervalvar length) added to figure captions for each taxa/functional group (Figures 1-7, now 1-8 with additional figure)

**Supplementary material – now included**

1. Table S1. Measurements of seawater conditions at time of sampling from Prydz Bay, Antarctica (19th November 2014).

2. Table S2. Mean carbonate chemistry speciation of DIC and $pH_T$ (measured) and $fCO_2$ and PA (calculated) for each minicosm tank after acclimation (days 8 to 18).

3. Figure S1. Temporal development of DIC within each minicosm throughout the experimental period.

4. Figure S2. Temporal development of $pH_T$ (total proton scale) within each minicosm throughout the experimental period.

5. Figure S3. Nitrate/nitrite ($NO_x$) concentrations within each minicosm throughout the experimental period.

6. Figure S4. Dissolved reactive phosphorus (P) concentrations within each minicosm throughout the experimental period.

7. Figure S5. Molybdate reactive silica (Si) concentrations within each minicosm throughout the experimental period.

**Anonymous Referee #1**

General comments This study investigates the effect of elevated $CO_2$ concentrations on protistan community composition of Prydz Bay, East Antarctica. As the quantification of cell abundances at the species level is very labour intensive, often studies tend to neglect this very important aspect that has been considered in detail in the study by Hancock and co-authors. The data presented are interesting and I belief it should be published, but it needs a considerable revision to be acceptable for Biogeosciences.

The authors need elaborate in more detail about the counting procedure, in particular for cells which were present only in low abundance as they often tend not to be evenly distributed in the Utermohl chamber, causing easily wrong cell abundance estimates.

The section "Methods – Light Microscopy" has been expanded to include more detail on the counting procedure and the steps taken to ensure representative abundance

estimates were gained for all taxa/functional groups. The Utermohl counting method described in Olrick *et al*. 1998 for protistan abundance estimates was used with a number of additional steps. In brief, a stratified counting procedure (small cells <20 μm at 400x and large cells >20 μm at 200x) was employed to provide both accurate identifications of small cells that are difficult to identify under 200x magnification, but still allowing accurate estimates of larger cells that have lower abundances (therefore fewer cells per field of view, FOV) at 400x magnification. To check that abundance estimates were accurate, mean cell counts of each taxon was recorded versus number of FOVs counted. These plots showed that the mean stabilised at 10-15 FOVs for small cells and 15-20 for large cells, therefore 20 randomly chosen FOVs were counted at each magnification. For nanoplanktonic cells (<20 μm), 20 randomly chosen FOVs at $3.66 \times 10^6 \mu m^2$ were counted providing on average counts totalling approximately 2,000 cells, ranging from 50 for rare taxa and over 1,000 for abundant species. For microplanktonic cells (>20 μm), 20 randomly chosen FOVs at $2.51 \times 10^5 \mu m^2$ were counted, providing on average counts of approximately 1,000 cells, ranging from 5 for rare taxa to over 300 for abundant chain forming taxa (i.e. *Chaetoceros*). Lastly, rare species with similar ecological function, and similar response to $f$CO$_2$ treatment, were combined together into functional groups to reduce noise in the multivariate analysis. These rare species with high standard deviation in their abundance estimate were identified in the results, and not discussed further in terms of response to $f$CO$_2$.

For better readability of the manuscript, information on carbonate chemistry as well as on macronutrient concentrations over the course of the experiments is needed.

Details on carbonate chemistry and macronutrient measurements are now provided in a supplement.

In particular, the onset of nutrient limitation on day 16 needs to be accounted for in the discussion of the development of the protistan community, which has been neglected so far. At the moment, the discussion mainly concentrates solely on the CO2 effects, which is fine until day 15, but not after this time point. This aspect needs to be addressed.

Throughout the results consideration has been included on the effects of $f$CO$_2$ on the abundance of taxa/functional groups and community structure prior to day 18 of the experiment, when nitrate/nitrite levels dropped below the level of detection. An additional paragraph within the section "Autotrophic protist taxa specific responses" has been included to address the potential effect of nutrient depletion on the autotrophic responses to $f$CO$_2$. Also, information has been added to the section "Community-level responses", I.e. NO$_x$ had no statistically significant effect on community structure in the multivariate analysis, as opposed to phosphorus and silicate (which were replete throughout the entire experiment).

For better and faster comparability of the figures of species-specific cell abundances, I recommend to use the unit 'cells per mL'.

Figures have been adjusted to show "cells per mL", rather than "cells per L" with an exponent.

Furthermore, to strengthen the author's argument that growth of large-sized diatoms is more prone to high CO2 concentrations, a graph showing actually the different trends in total abundance of all small versus all large diatoms, similar to figure 3, is needed.

A figure (Figure 2 in the amended manuscript) similar to figure 3 has been added as suggested. This plot clearly shows the different responses of large (>20 µm) and small (<20 µm) diatoms to $f$CO$_2$ treatment.

Introduction P2, L5-6: This statement is not right as there are several studies that were already published on OA effects in various natural assemblages of Southern Ocean microbes (Tortell et al.. 2008, Feng et al.. 2010, Hoppe et al.. 2013, McMinn et al.. 2014, Young et al.. 2015, Coad et al.. 2016, Davidson et al.. 2016, Thomson et al.. 2016). Please rephrase.

This sentence has been removed.

P2, L19-35: Considering that the authors already cited 8 papers that were published on CO2 effects, it is not really appropriate to write that "there have been relatively few studies". Please also cite the studies by Hoppe et al.. 2013 PLOS One and Young et al.. 2015 MEPS, which are currently missing. The latter two studies need also to be taken into account when summarizing the findings on CO2-dependent shifts in community composition in this paragraph.

This sentence has been removed and the whole paragraph re-written to accurately summarise the findings of previous studies including Hoppe *et al.* 2015 and Young *et al.* 2015.

P2, L23-25: Please note that Feng et al.. (2010) reported a shift from Cylindrotheca to Chaetoceros from 380 to 750 $\mu$atm pCO2, and not from Pseudo-nitzschia. Further Tortell et al.. (2008) did not observe a CO2-triggered shift in Phaeocystis antarctica. It was reported that both summer and spring phytoplankton communities were dominated by P. antarctica and within the communities a shift among diatoms was observed.

These sentences have now been re-written to accurately summarise the findings of Feng *et al.* 2010 and Tortell *et al.* 2008, and the species covered in those papers.

Methods P3, L17: Did the authors assess whether the gravity filtration procedure introduced cell damage and/or physiological fitness of the sampled microbial community? The latter could have affected the evolution of the community structure.

There was no direct assessment of the potential effects of gravity filtration on cellular fitness, however the filling speed was slow to prevent damage due to turbulence (this is now added to the filling description in methods). Furthermore, we would not expect negative impacts on the cells that are small enough to pass through the mesh into the mesocosms.

P3, L25: To me is unclear why during the initial acclimation phase the community was exposed to the extremely low light intensity of âĹij1 $\mu$mol photons m-2 s-1.

[Figure]

A low light intensity was used during the acclimation period to preclude growth of the phytoplankton community whilst cellular physiology acclimated to the increase in $f\mathrm{CO}_2$ to target levels in each minicosm.

This information is now included in the methods section that has been restructured (outlined below).

P3, L28-32: How was the light intensity adjusted? Were the minicosms not exposed to the natural irradiance cycle? Did the authors monitor daily in situ irradiances over the whole experiment? The manipulation of the light intensity remains unclear to me.

The beginning of the methods section has been restructured to include a "Minicosm operation" section.

The minicosms were not exposed to a natural irradiance cycle as the minicosms were contained within a single shipping container with artificial lighting. The light intensity and cycle during the incubation phase of the experiment (days 8 to 18) was saturating but not inhibitory to the phytoplankton.

P4, L2-12: I can understand that carbonate chemistry results are reported in detail in Deppeler et al.. (submitted), but also for this manuscript there is the need to give information on the successful CO2 manipulation of each CO2 treatment at least in a table. For the interpretation and discussion on the results of the development of the community composition, it would be also helpful to give the information on carbonate chemistry (e.g. pH, fCO2) at the day of seawater sampling.

Carbonate chemistry speciation throughout the experiment and at the time of seawater collection is now provided in supplementary material.

P5, L2-11: The counting of particularly large diatoms can be problematic. To this end, it is recommended to count the whole Utermöhl chamber as species are not distributed evenly. In particular, chain-forming diatom species can be very patchy, making their quantification on the basis of 20 chosen fields of view difficult. How many cells did the

authors count per species? How was the patchiness of species distribution within the chamber accounted for? Considering the low cell numbers, it is important to address this issue as otherwise easily wrong cell abundance estimates can be made.

The section "Methods – Light Microscopy" has been expanded to include a more detailed description of the counting procedure. A brief description of the extra steps taken to assure accurate estimates is mentioned above.

I miss information on the development of the macronutrient (N, P, Si) concentrations over the duration of the experiment. This info needs to be provided either in a table or a figure.

Macronutrient concentration changes over time are now provided in a supplement.

According to Deppeler et al.. (submitted to Biogeosciences) N was depleted for most treatments at day 16, this means that in addition to the changes in fCO2 N also potentially acted as stressor at the end of the experiment, potentially influencing community composition at the end of the experiment. The latter information is not obvious when nutrient data are not shown in this manuscript and needs to be accounted for in the interpretation and discussion of the results. Hence, to assess the effect of increasing fCO2 levels on community composition, the authors should rather compare results at day 16 instead of day 18. For instance, the abundance of Fragilariopsis species < 20 $\mu$m of the 343, 506, and 634 $\mu$atm fCO2 treatments strongly dropped between day 16 and 18, coinciding with nitrate limitation at these specific fCO2 levels.

Nitrate/nitrite fell below the level of detection only at day 18 of the experiment ($NO_x$ measurements throughout the experiment are now shown in a supplement). A consideration of single-species and community structure responses to $fCO_2$ prior to day 18 has now been included in the results and discussion. Concerning the decrease in nano-sized (<20 µm) Fragilariopsis abundance between day 16 and 18 of the experiment, the low $NO_x$ concentrations on day 16 do not correlate with a decline in cellular abundances on day 18 as, for instance, there was hardly any change in abundance

between these two days at 953 μatm despite relatively low $NO_x$ levels. This is also observed in other taxa i.e. *Chaetoceros*.

Results

P6, L21-22: It seems very unlikely that the high variation in protist abundance of the 635 $\mu$atm pCO2 treatment, accounting for âĹij10000 cells per mL, comes from the increase in rare large cell diatom species, which were only present between 5 up to 200 cells per mL (Fig. 2-5).

The observation of anonymous reviewer 1 is valid; therefore this sentence has been removed.

P6, L27-29: It is not really clear which figure underlines this statement. Also, it would be helpful to point out which diatoms were classified as 'large' and 'small'. For this statement, it would be good to have a graph showing actually the different trends in total abundance of all small versus all large diatoms, similar to figure 3.

A new figure has been added to the manuscript showing large (>20 μm) vs. small (<20 μm) diatom abundances.

P6, L30: Do the authors now refer to Fig. 2b-d when they refer to discoid centric diatoms or is Fig. 2a meant, but then it does not make sense to refer to 'unidentified discoid centric'. Does the latter term refer to one single species or does it summarize all counts of unidentified discoid diatom species that were smaller than 2 $\mu$m? Another idea would be to add the cell volume of the species next to its name on the graph, making it easier to see the size differences at a glance.

The term "discoid centric diatoms" has been clarified by including a description in what the term is referring to (centric diatoms with a valvar diameter greater than the pervalvar dimension i.e. centric diatoms of the genera Thalassiosira, Landeria and Stellarima). The wording "unidentified discoid centric" has also been amended to reflect that this is referring to all discoid centric diatoms with a valve diameter <2 μm which could not be

identified to genus or species level. The average longest length dimension has been added to all $f\mathrm{CO_2}$ response graph captions (Figures 1 to 5, now 1 to 6 with additional large vs. small figure). We believe that adding it into the name of the graph will clutter the figure and make the graphs difficult to read; therefore added it into the caption. We have used the average longest length dimension as the sizing parameter as this has already been used throughout the manuscript i.e. Fragilariopsis below and above 20 µm in length (Figure 3 and P7L10-17).

P7, L14: To what fCO2 treatment does the control refer to? 343 $\mu$atm?

We removed "control" and replaced it with "ambient (343 µatm)" throughout the manuscript.

P7, L12-14: Looking at figure 3a, small Fragilariopsis cells of the 953 $\mu$atm reached highest cell abundances in comparison with all other fCO2 treatments at day 16 and 18. The authors, however, write "Abundances in the ÆŠCO2 treatments >953 $\mu$atm were lower but less than those in the control treatment. . ." How can this be?

The wording was adjusted to correctly reflect the response seen in Figure 3a.

P7, L15-17: Why is a tolerance lower when cell abundance is higher?

The sentence was amended and wording made clearer.

P7, L20-21: The species name is O. weissflogii and not weissfloggi. Also write 'pennate' instead of 'pennant'. Also, it is Pseudo- nitzschia and not Pseudonitzschia. Also change turgidulodies to turgiduloides.

The miss-spelling was amended.

Discussion

P9, L26-28: As in almost all figures cell abundances did not change between day 1 and 8, considering also that irradiance was very low, I am surprised about the statement that community composition changed.

Additional information has been added to the discussion section "Acclimation to high $CO_2$" to clear confusion regarding the change in community composition during the acclimation period of the experiment. During the acclimation the light irradiance was very low to preclude phytoplankton growth during physiological acclimation, therefore this change in community composition is hypothesized to be due to death of delicate cells (from sample collection or the minicosm environment ie. lighting, being sub-optimal and different species having different environmental requirements/sensitivities). This change in community composition was the same across all minicosm tanks/ $f\text{CO}_2$ treatments (as shown in the cluster analysis, Figure 9).

Did the authors characterize species composition of the initial community? As the information on the characterization of the initial community is missing this complicates the interpretation on subsequent species changes through the sampling.

The initial community was characterized on day 1 of the experiment, one day after filling of the tanks and prior to any $CO_2$ manipulation (outlined in the Methods section).

P9, L 32-33: Taking into account the very low irradiance between day 1 and 8 the cells were exposed to, it is not surprising the community showed a severe delay in growth among all treatments, a finding which s not mentioned here. Apparently, the combination of very low irradiance and high fCO2 caused even stronger delay. This is worth to be mentioned. On which observation is the statement based that 'the protists required more than 8 days to acclimate to this high fCO2'?

After careful calculations, there is no "lag in growth" as described in the manuscript. As such we have removed this finding from the discussion and conclusion.

P10, L1-6: To underline the statement that community growth of the highest fCO2 treatment was lowest, why do the authors not calculate community growth rate? All data are there and this would strengthen their argumentation.

As described above this has now been removed from the manuscript.

P10, L9-15: I am not yet convinced about the statement that in 'diatoms the response was mainly size-related'. To underline this, a graph showing actually the different trends in total abundance of all small versus all large diatoms, similar to figure 3, is required. The authors even point out that 'a couple of species did not follow this trend'.

An additional figure has been included with small versus large diatom cells as described above. We have now reworded this section in the discussion to reflect that whilst the majority of the diatom species follow this size-related trend, there is a few that do not (i.e. *Proboscia* and *Chaetoceros*).

P10, L9-19: The fact that nutrients became limiting either on day 16 or 18 needs to be elaborated in more detail. This aspect was fully neglected, only in L14 it is mentioned that '*Chaetoceros* did not show a response to fCO2, but instead reflected the nutrient availability'. This aspect needs to be discussed also for the other species.

Done.

P10, L19-20: The low tolerance to high pCO2 is also found and reported in Tortell et al.. (2008) and Hoppe et al.. (2013).

A comparison to Tortell *et al.*, 2008 and Hoppe *et al.*, 2013 has been added.

P10, L20: 'Unlike diatom species, . . . Phaeocystis dramatically declined . . . at the three highest fCO2 levels'. It was, however, pointed out before that 'large diatoms showed . . .a decrease at higher fCO2'. There is no controversy, please modify.

The sentence has been amended to reflect that whilst *Phaeocystis* is a smaller cell, its response to $f$CO$_2$ is similar to that of larger not smaller diatoms.

P10, L23: Please specify the statement 'our study only finds this response in diatoms'. To which response is referred to? The increase in diatom abundance under high fCO2? But the opposite response for diatoms was claimed before.

Paragraph re-written.

P10, L21: I disagree that there is a 'common consensus in other ocean acidification studies that pico- and nanoplankton abundance increases at high CO2 levels'. Like the dataset of the authors, there are several studies reporting the opposite for Southern Ocean com- munities (Tortell et al.. 2008, Feng et al.. 2010, Hoppe et al.. 2013). Please rephrase more carefully.

Sentence removed.

P10, L23-25: Repetition, please see L20.

Lines 17-19 removed.

P10, L25-28: In line with the data by Hancock et al.., in none of the cited studies Phaeocystis antarctica showed a positive growth response to high CO2, growth rather remained unaffected by CO2. Please also add Trimborn et al.. 2017 Physiol. Plant, which is in line with the latter observation.

Sentence adjusted and Trimborn *et al*. 2017 reference added.

P10, L28-30: I disagree. The results from Feng et al.. (2010) show no CO2 effect on the colonial Phaeocystis antarctica. Furthermore, it is not clear to me why the presence of the mucus could have any effect on the CO2 sensitivity of Phaeocystis.

These sentences have been removed in the re-writing of the discussion section "Au- totrophic protist taxa specific responses".

P11, L1: Please also cite Wu et al.. 2014 that reported enhanced growth rates in response to high CO2 in large diatoms.

The Wu *et al*. 2014 citation added.

P11, L3-6: As mentioned before, no CO2- dependent increase in Phaeocystis was re- ported in Tortell et al.. (2008), Feng et al.. (2010) or Trimborn et al.. (2013). Please correct. Further Xu et al.. did also not observe any CO2-dependent increase in Phaeo- cystis from the current to the 2060 scenario, but a significant decline from 2060 to 2100.

Please note that in the latter study next to CO2, also temperature, light and Fe availability was changed, being therefore more difficult to compare with this data set here.

Sentences removed.

P11, L13-15: Please also cite Trimborn et al.. (2013) who actually investigated the CCM of Southern Ocean phytoplankton species, among them Phaeocystis antarctica.

The findings from Trimborn *et al*. 2013 have now been included.

P11, L26-35: For better readability, please specify the direction of the observed responses of the different choanoflagellates, just saying 'there were differences' is not enough.

Done.

Discussion Part 4.4: In particular here, the onset in nutrient limitation at day 16 and 18 needs to be accounted for in the discussion of community-level responses as CO2 was not the only driver. The latter statement also applies for the overall discussion.

Additional discussion has been added to "Community-level responses" considering the depletion of nutrients during the experiment on the protistan community structure and succession (in which $NO_x$ was not significant in the multivariate analysis only phosphorus and silicate which were replete throughout the entire experiment). As described above this has also been included in the "Autotrophic protist taxa specific responses" section of the discussion.

Figures: Fig. 1-7: For better and faster comparability between cell abundances of the different species, I would use the unit 'cells per mL' in all figures instead of using 'cells x 104L-1'asinFig. 2-4,'cellsx105L-1'asinFig. 7or'cellsx107L-1'asinFig. 1and6. The latter makes it even more complicated as the Y-axis is also changing, hampering a fast comparison between cell numbers between graphs of different figures.

Figures have been adjusted to cells per mL.

In the leg ends of Fig. 1 to 7, it is referred to the pCO2 while in the MM section it is referred to fCO2, please stick to one of them throughout the manuscript. pCO2 adjusted to fCO2.

Legends have been adjusted within Figures 1-7 to $f\mathrm{CO}_2$ rather than $\mathrm{pCO}_2$.

**Anonymous Referee #2**

This manuscript analyses the effects of elevated CO2 on the protistan community in East Antarctica. Firstly it is great to read another biological ocean acidification study being conducted in the Antarctic as well as being a community response study. Both these areas of research are not common with many questions left unanswered. It is, therefore, particularly interesting that this study by Hancock et al.. addresses community level responses in the Antarctic where biota are considered to be the most vulnerable to OA due to the rising solubility of CO2 in cold-waters. Overall this manuscript is well written, contains plentiful relevant data and attempts to close the gaps in our knowledge of important questions outstanding in the OA field.

Before consideration for publication, there are a few points that need addressing. In particular, more explanation is needed about the carbonate chemistry analysis.

The section of the methods addressing carbonate chemistry has been expanded. In addition the carbonate chemistry speciation throughout the experiment and at the time of seawater collection has been included into the supplementary material (Table S1 and S2, and Figure S1 and S2).

DIC and pHT are the CO2 parameters directly measured so why is fugacity of CO2 used as the CO2 parameter altered? pH (on any scale) or DIC are the usual parameters directly altered in OA studies and therefore using fCO2 limits the continuity between this study and others. I suggest using either the measured pH or DIC measurements instead.

Like many other studies we have manipulated carbonate chemistry by increasing DIC

and leaving total alkalinity constant which perfectly mimics ongoing ocean acidification. The data is presented as $f\mathrm{CO_2}$ as this is preferred rather than pH or DIC as $f\mathrm{CO_2}$ corresponds to a certain time in the future based on emission scenarios. That is the reason why most studies opt to discuss their results with respect to $f\mathrm{CO_2}$ rather than DIC or pH.

A table of differences between the carbonate chemistry of each treatment is also necessary rather than quoting Depper et al. (submitted).

Table S1. Measurements of seawater conditions at time of sampling from Prydz Bay, Antarctica (19th November 2014).

Table S2. Mean carbonate chemistry speciation of DIC and $\mathrm{pH_T}$ (measured) and $f\mathrm{CO_2}$ and PA (calculated) for each minicosm tank after acclimation (days 8 to 18).

Figure S1. Temporal development of DIC within each minicosm throughout the experimental period.

Figure S2. Temporal development of $\mathrm{pH_T}$ (total proton scale) within each minicosm throughout the experimental period.

Many more details are generally needed. For example, how often was DIC measured? What was the variability between measurements in DIC and pH? How often was the probe calibrated?

The section of the methods addressing carbonate chemistry has been expanded. Daily DIC and $\mathrm{pH_T}$ measurements throughout the experiment are now included in the supplementary material Figures S1 and S2. The pH probe was only calibrated initially with freshwater buffers as it was simply used as an indicator of how much $\mathrm{CO_2}$ enriched seawater had to be added to the minicosm to maintain the $\mathrm{CO_2}$ level. Actual carbonate chemistry speciation samples were taken and measured after the addition using more suitable and robust methods (DIC, spectrophotometric $\mathrm{pH_T}$).

A common problem with OA research is replication. I query why this experiment was

not replicated given the short duration? In addition OA research is also moving towards long-term studies spanning many months to years. I also query why such a short duration was chosen for this experiment?

The experiment was only conducted once due to having a short time period available for set-up, running of the experiment and pack-up between transport options to and from Davis Station (ship and flights).

Throughout the manuscript there are several references to look at Deppler et al. (submitted) for information not detailed in this manuscript. I query whether this manuscript is a "stand-alone" story.

Whilst this manuscript is complementary to Deppeler *et al.*, it is a "stand-alone" story. Our manuscript reports single species responses, differences between species and changes in protistan community composition. In contrast, Deppeler *et al.* (submitted) presents the photo-physiological responses of the phytoplankton community. Carbonate chemistry and macronutrient data has now been added to our manuscript and, therefore, the number of references to Deppeler *et al.* is now minimal.

Technical corrections

Page 1 line 1: remove 'of'

'Of' was removed.

Page 1 line 8: should be a semicolon instead of a colon.

Colon was changed to semicolon.

Page 1 line 8/9: is it a case of large cells decrease in abundance in high fCO2? That would be a better way to report these results, as high fCO2 is the environmental stress concerned.

Sentence was reworded.

Page 1 line 12/13: This statement needs clarification as it implies this research is not original.

Wording adjusted to; "Despite interannual differences and the time in the season which the experiment was performed, comparisons with previous experiments show that the threshold $f\mathrm{CO}_2$ remains the same for this nearshore site".

Page 2 line 4-6: generally OA studies on organisms higher up the foodweb in the Antarctic are few which adds importance to your study and should be mentioned with some key Antarctic papers referenced.

Additional discussion in the paragraph has been added describing indirect effects on higher trophic levels due to the phytoplankton community structural change.

Page 2 line 23: insert a comma. . .'With increased COÂnËĞ2, Tortell et al.....'

Paragraph re-written.

Page 4 line 5: remove 'the' before adding

Paragraph re-written.

Page 6 line 22: remove 'a' should be 'likely due to'

Sentence now removed through editing for comments by Anonymous Referee #1.

Page 7 line 3: change to 'had increased to'

Sentence now removed.

Results section: removed 'show' and other variations using this word as it is unnecessary. It reads better to just say 'increased' instead of 'showed an increase'.

The use of "show" removed throughout the results and discussion section of the manuscript.

Page 10 line 11: typo 'a' should be 'at'

'A' changed to 'at'.

Page 10 line 17-19: why might there be differences between the results in this study and that of Feng et al.. (2009)?

Section re-written to include comments by Anonymous Referee #1. Feng *et al*. 2009 citation now removed.

Page 10 line 22: it is difficult to compare the results in this study to others quoted in this statement when different CO2 parameters were altered.

This sentence has now been removed.

Page 10 line 23: 'response' instead of 'responses'

This sentence has now been removed.